# Bounded Hyperbolic Tangent: A Stable and Efficient Alternative to Pre-Layer Normalization in Large Language Models

## Abstract

Pre-Layer Normalization (Pre-LN) is the de facto choice for large language models (LLMs) and is crucial for stable pretraining and effective transfer learning. However, LN and Pre-LN are inefficient due to repeated statistical calculations and suffer from the curse of depth. As layers grow, the magnitude and variance of the hidden state escalate, destabilizing training. Efficiency-oriented, normalization-free methods such as Dynamic Tanh (DyT) improve speed but remain fragile at depth. To jointly address stability and efficiency, we propose Bounded Hyperbolic Tanh (BHyT), a drop-in replacement for Pre-LN. BHyT couples a tanh nonlinearity with explicit, data-driven input bounding to keep activations within a non-saturating range. It prevents depth-wise growth in activation magnitude and variance and comes with a theoretical stability guarantee. For efficiency, BHyT computes exact statistics once per block and replaces a second normalization with a lightweight variance approximation, enhancing efficiency. Empirically, BHyT achieves improved stability and efficiency in pretraining, delivering on average 7.7% faster forward computation and up to 5% higher token generation throughput than RMSNorm, while matching or surpassing its inference performance and robustness across language understanding and reasoning benchmarks. Our code is available at: `https://anonymous.4open.science/r/BHyT`

## 1 Introduction

The remarkable progress of large language models (LLMs) is closely tied to the Transformer architecture, in which normalization layers play a central role (Vaswani et al., 2017). In particular, Pre-Layer Normalization (Pre-LN), which applies normalization before each sublayer, has become the de facto standard (Xiong et al., 2020; Radford et al., 2019; Brown et al., 2020; Touvron et al., 2023). This design stabilizes optimization in very deep networks, enabling training at scale and strong transfer performance across diverse tasks. By regulating the scale and distribution of activations, Pre-LN provides a reliable training regime that helps unlock the modern era of large-scale LLMs (Ba et al., 2016; Xiong et al., 2020).

As models push to greater depth, however, these foundation models expose a critical weakness. Recent analyses identify a curse of depth, in which the interaction between residual connections and Pre-LN drives rapid, often near-exponential, growth in hidden-state magnitude and variance with layer depth (Sun et al., 2025). This rapid increase in hidden-state norms—often termed the massive activation phenomenon (Sun et al., 2024; Lin et al., 2024)—can induce strong biases toward particular tokens, which in turn degrades the model's generalization performance and makes it difficult to realize the benefits that increased depth should provide (Kim et al., 2025). Moreover, variance escalation is more important than simple divergence: it drives the block Jacobian toward the identity, so deep layers learn little beyond an expensive identity mapping (Sun et al., 2025). The core challenge is thus not merely preventing crashes; it is preserving informative transformations and efficient signal propagation so that every layer contributes meaningfully to learning.

The community has explored remedies along a stability–efficiency axis. On the stability end, stronger normalization seeks to suppress depth-wise drift. For instance, Peri-LN applies normalization both before and after each sublayer, curbing variance growth and maintaining gradient health

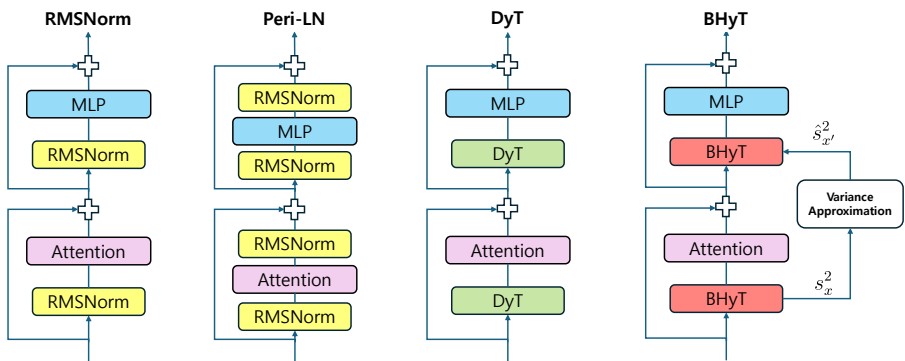

Figure 1: Architectural comparison of normalization strategies in Transformer blocks. **RMSNorm** applies normalization before each sublayer to stabilize activations but suffers from variance growth at scale. **Peri-LN** reinforces stability by normalizing both before and after each sublayer. **DyT** replaces normalization with a lightweight scaled tanh nonlinearity with learnable scalar $\alpha_{\text{DyT}}$. **BHyT** (ours) combines bounded tanh with data-driven variance control: it computes input variance once per block, approximates subsequent variance for efficiency, and explicitly constrains activations to a non-saturating range, thereby unifying stability and efficiency.

even at extreme depths (Kim et al., 2025). The trade-off is substantial overhead: doubled normalization increases reduction operations and memory traffic, inflating latency. On the efficiency end, RMSNorm removes mean centering to reduce computational costs and has been widely adopted in modern LLMs (Zhang & Sennrich, 2019). Yet it only partially controls activation statistics and does not fundamentally resolve variance accumulation, leaving depth scaling fragile.

A more radical line of work questions the need for normalization at all. Dynamic Tanh (DyT) replaces normalization with a simple element-wise nonlinearity and a learnable scale (Zhu et al., 2025). By eliminating per-token statistics, DyT accelerates both training and inference. However, this efficiency comes with a major risk: activation saturation. When inputs grow large, precisely the hazard under depth, the nonlinearity saturates, gradients vanish, and optimization becomes brittle. Relying on a learned global scale to self-regulate these dynamics is often insufficient in the face of strong depth-induced amplification.

This landscape reveals a clear gap: methods that guarantee stability typically sacrifice efficiency, while efficient methods fail to robustly address the curse of depth. What is needed is a mechanism that retains the simplicity and throughput of normalization-free designs while providing principled control over depth-wise growth.

We introduce Bounded Hyperbolic Tanh (BHyT) to close this gap. As shown in Figure 1, BHyT is a drop-in replacement for Pre-LN that couples a tanh nonlinearity with explicit, data-driven input bounding. By keeping activations within a non-saturating regime with high probability, BHyT prevents depth-wise growth in activation magnitude and variance, thereby preserving informative transformations as depth increases. Beyond this forward-path protection, BHyT comes with a theoretical stability guarantee: its gradient scale is provably upper-bounded by that of RMSNorm, ensuring at least comparable stability even in very deep stacks. We further show theoretically that, under realistic conditions, BHyT can attain strictly lower layerwise variance than the variance-constraining method proposed in prior curse-of-depth work Sun et al. (2025). To minimize overhead, BHyT computes exact statistics once per block and replaces a second normalization with a lightweight variance approximation that tracks attention-induced variation. This variance approximation is computed in parallel with the main path of the transformer block (sequential attention and MLP modules) during training, thereby minimizing additional latency. Empirically, we observe that BHyT is about 7.7% faster than RMSNorm in the forward pass and achieves roughly a 10% speedup during training, while largely preserving or even slightly improving language reasoning benchmark performance compared to RMSNorm or other baselines.

Our contributions are summarized as follows.

- We propose BHyT, coupling bounded nonlinearity with a parallelizable variance approximation. This eliminates redundant statistical reductions, effectively mitigating the curse of depth.

- We provide a rigorous stability analysis, including the finite-depth variance bound, proving that BHyT achieves strictly lower variance growth than the baselines under realistic conditions.

- We validate BHyT on Llama-1B/3B architectures; our experiments, covering both depth-wise stability and inference speed, demonstrate superior accuracy and a $\sim 12.9\%$ increase in inference throughput over Peri-LN.

## 2 RELATED WORKS

### 2.1 LIMITATIONS OF PRE-LAYER NORMALIZATION

Layer normalization has been a key component in Transformer architectures (Ba et al., 2016), and its Pre-LN variant has become the default in most LLMs (Xiong et al., 2020). The Pre-LN design, which normalizes hidden states before attention and feed-forward layers, improves stability when training starts. Still, this design is not without drawbacks. From a stability perspective, recent studies have shown that Pre-LN suffers from massive activations as depth increases (Sun et al., 2025; Kim et al., 2025). In this regime, the mean and variance of hidden states grow rapidly across layers, destabilizing optimization. From an efficiency perspective, normalization requires computing statistics such as the mean and variance of activations. These reductions introduce latency and memory overhead, which accumulate as depth increases. As a result, Pre-LN can both destabilize very deep models and become a bottleneck for training throughput and scalability.

### 2.2 NORMALIZATION-BASED ENHANCEMENTS

The main bottleneck of normalization-based designs lies in the repeated computation of statistics at every block. A first attempt to reduce this cost was RMSNorm, which discards the mean and normalizes only by the root mean square of activations (Zhang & Sennrich, 2019). This removes part of the overhead while keeping activations within a reasonable range. However, variance computation is still required for every layer, and the instability of Pre-LN remains unresolved.

To address this instability, LayerNorm Scaling (LNS) (Sun et al., 2025) introduces a layer-index scaling that constrains variance growth across depth. By rescaling activations according to layer index, it alleviates the massive activation problem and stabilizes optimization in very deep networks. Although LNS improves stability, its reliance on per-block statistics keeps the cost high in large-scale training. Peri-LN reinforces stability further by applying normalization both before and after each sublayer (Kim et al., 2025). This suppresses variance spikes and improves convergence and downstream accuracy; however, doubling normalization operations adds substantial overhead, making it less suitable for large-scale pretraining.

### 2.3 NORMALIZATION-FREE ALTERNATIVES

To avoid the overhead of statistical computations altogether, recent work has explored removing normalization. DyT follows this direction by replacing RMSNorm with a bounded activation function, $\tanh(\alpha_{\text{DyT}}x)$, where $\alpha_{\text{DyT}}$ is a learnable scalar (Zhu et al., 2025). This eliminates the need for mean or variance estimation, improving both training and inference speeds. The $\tanh$ function ensures that activations remain bounded, and the learnable scalar $\alpha_{\text{DyT}}$ rescales inputs so that the effective range of $\tanh$ is better utilized.

Although DyT improves efficiency, it does not control how the output mean and variance grow with depth. Moreover, its learnable scalar paramter $\alpha_{\text{DyT}}$ is not explicitly designed for stability. It only rescales the inputs indirectly, without actively suppressing variance escalation in deep networks. As a result, its training stability at scale is not explicitly guaranteed and may be more susceptible to saturation-induced gradient vanishing and related instabilities than normalization-based methods.

In contrast to DyT, BHyT explicitly guarantees stability by constraining variance growth. To address the computational bottleneck of the required statistical calculations, we introduce a variance approximation mechanism. This design effectively harmonizes robust training stability with the high efficiency of normalization-free architectures.

## 3 METHODOLOGY

An ideal normalization scheme should allow for controlled, sub-exponential or linear growth of variance, sufficient to preserve signal propagation without leading to explosive instability. To this end, we introduce BHyT, a novel normalization layer designed to achieve both stability and speed. BHyT builds upon the efficiency of methods like DyT, which use S-shaped functions (e.g., $\tanh$) to bypass explicit statistical computation. However, DyT's reliance on a single learnable scalar to control the input to $\tanh$ offers no explicit guarantee against saturation, which can lead to vanishing gradients in very deep models. BHyT addresses this limitation by explicitly bounding the input to the non-linear function, ensuring both numerical stability and a well-behaved gradient flow.

### 3.1 ENSURING STABILITY VIA INPUT BOUNDING

To prevent the saturation of the $\tanh$ function and ensure stable gradient flow, BHyT explicitly constrains its input to lie within a predefined range $(-\lambda, \lambda)$ with high probability. We first define a formulation, BHyT*, that utilizes input statistics to achieve this bound. Given an input $x \in \mathbb{R}^d$ with a mean $\mu_x$ and standard deviation $s_x$, BHyT* is defined as:

$$\text{BHyT}^*(x) = \gamma \odot \tanh\left(\frac{\lambda}{\kappa s_x + |\mu_x|} x\right) \tag{1}$$

where $\gamma \in \mathbb{R}^d$ is a learnable scale parameter, $\odot$ denotes the element-wise product, $\lambda > 0$ is a hyperparameter defining the range of the bound, and $\kappa = (1-p)^{-1/2}$ is derived from a target probability $p \in (0,1)$ as shown in Proposition 3.1.

**Proposition 3.1** (Input scaling bound at the level $p$). *Let $x \in \mathbb{R}$ with $\mathbb{E}[x] = \mu_x$ and $\text{Var}(x) = s_x^2$. Given a magnitude budget $\lambda > 0$ and target probability $p \in (0,1)$, if $|\alpha| = \frac{\lambda}{\kappa s_x + |\mu_x|}$, then $\mathbb{P}(|\alpha x| \leq \lambda) \geq p$ for any distribution with finite variance, where $\kappa := (1-p)^{-1/2}$.*

**Remark 3.2.** Without loss of generality, we set $\alpha \geq 0$ since only $|\alpha|$ matters.

**Remark 3.3.** For $p = 0.99$, $\kappa = 10$ and $|\alpha| = \lambda/(|\mu_x| + 10s_x)$.

This bounded formulation provides strong theoretical guarantees regarding stability. As shown in Theorem 3.4, the Jacobian norm of BHyT is upper-bounded by that of RMSNorm.

**Theorem 3.4** (Jacobian norm bound). *Let $J_{BHyT^*}(x)$ and $J_{RMS}(x)$ be the Jacobians of BHyT* and RMSNorm, respectively. The denominator for the input of BHyT* is $\phi(x) = \kappa\sqrt{s_x^2 + \varepsilon} + |mu_x|$, while for the input of RMSNorm it is $s(x) = \sqrt{\mathbb{E}[x^2] + \varepsilon}$. Under the assumption of zero-mean inputs ($\mu_x = 0$), we have $\phi(x) \approx \kappa \cdot s(x)$, and it induces:*

$$\|J_{BHyT^*}(x)\|_2 \leq \frac{\lambda}{\kappa} \|J_{RMS}(x)\|_2.$$

This result guarantees that the Jacobian norm of BHyT* is upper-bounded by RMSNorm. This implies that our method offers superior training stability with respect to gradient magnitudes.

### 3.2 ACCELERATING BHyT WITH VARIANCE APPROXIMATION

A key strategy for accelerating Transformer models is to replace standard LN (e.g., RMSNorm) with a $\tanh$ function. This approach can significantly improve computational speed by eliminating the need for explicit statistical calculations inherent in layers like RMSNorm. However, BHyT*, which requires the per-instance calculation of mean and variance, would lose this speed advantage. Motivated by this, we introduce BHyT, an approximate variant of BHyT*. Although closely related to BHyT*, BHyT employs variance approximation, thereby enhancing both stability and efficiency.

Our solution, BHyT, successfully navigates this trade-off by approximating the variance instead of computing it directly. To illustrate its efficiency, while a standard Transformer block computes statistics twice (e.g., via RMSNorm), BHyT requires only a single direct variance calculation at the entrance of the block. All subsequent variances for intermediate states are approximated. This

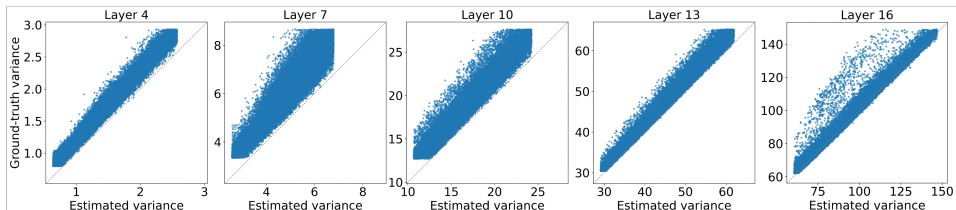

Figure 2: Approximated and ground-truth activation variances for the second BHyT$_{\text{mlp}}$ layer across different transformer blocks in Llama-1B. The variance estimates are evaluated using 100 randomly sampled inputs from the C4 training corpus. The diagonal line represents the ideal $y = x$ reference. Deeper layers exhibit closer alignment with this reference, indicating improved approximation accuracy. Complete results for all 16 layers are reported in the Appendix D.3.

drastic reduction in statistical overhead leads to substantial speed improvements. While BHyT fundamentally assumes a zero-mean input (similar to RMSNorm), for added notational simplicity in this section, we also omit the $\varepsilon$ term.

### 3.2.1 BHYT BEFORE THE ATTENTION LAYER

For the first normalization within a block, BHyT directly computes the variance of the input tensor $x$ across the feature dimension. The BHyT is then applied as follows:

$$z_{\text{attn}} = \text{BHyT}_{\text{attn}}(x) = \tanh(\frac{\lambda}{\kappa\sqrt{s_x^2 + \varepsilon}}x) \tag{2}$$

where $s_x^2$ denotes the per-instance variance calculation, and $\varepsilon$ denotes a tiny constant added to avoid numerical instability. The output $z_{\text{attn}}$ is then passed to the self-attention layer, $\mathcal{A}(\cdot)$, which produces the attention output $h_{\text{attn}} = \mathcal{A}(z_{\text{attn}})$.

### 3.2.2 BHYT BEFORE THE MLP LAYER

Let $x' = x + h_{\text{attn}}$ be the output of the first residual connection and the input to the second BHyT (i.e., BHyT$_{\text{mlp}}$). $h_{\text{attn}}$ denotes the output of the self-attention layer. Instead of re-computing the variance of $x'$, which is computationally expensive, we approximate it as

$$\hat{s}_{x'}^2 \approx s_x^2 + \tilde{s}_{h_{attn}}^2. \tag{3}$$

As shown in Equation 3, our approximation for the intermediate variance, $\hat{s}_{x'}^2$, is the sum of two components. The first term, $s_x^2$, is the variance of the Transformer block's input, reused directly from the preceding BHyT$_{\text{attn}}$ computation. The second term, $\tilde{s}_{h_{\text{attn}}}^2$, is our efficient approximation of the variance contributed by the attention layer's output. Similarly to Equation 2, BHyT$_{\text{mlp}}$ then utilizes this approximated variance. It allows BHyT to maintain normalization stability while minimizing the overhead from repeated statistical calculations within a single Transformer block.

**Theorem 3.5** (Variance approximation of attention output). *Let $x \in \mathbb{R}^{T \times d}$ be the input to a self-attention layer, where $T$ is the sequence length and $d$ is the hidden-state dimension. The attention layer has value and output projections $W_V \in \mathbb{R}^{d \times d_V}$ and $W_O \in \mathbb{R}^{d_V \times d}$. Assume that, for a large sequence length $T$, the attention weights are approximately uniform, and that the variance $\frac{\lambda^2}{\kappa^2}$ is sufficiently small to justify the linearization of the $\tanh$ function. The variance of the attention layer's output, $\tilde{s}_{h_{attn}}^2$, can then be approximated as:*

$$\tilde{s}_{h_{attn}}^2 \approx \frac{1}{Td}\|W_V W_O\|_F^2 \cdot \frac{\lambda^2}{\kappa^2}$$

Theorem 3.5 demonstrates that the variance induced by the attention module can be approximated solely using model characteristics—such as the attention layer weights, sequence length, and BHyT parameters—independent of input values. The detailed proof is provided in Appendix B.3. This independence allows the variance approximation to be computed in parallel with the forward path,

---

**Algorithm 1** Transformer Decoder Block with BHyT

---

1: **Input:** $x \in \mathbb{R}^{T \times d}, W_V, W_O, \gamma_{\text{attn}}, \gamma_{\text{mlp}} \in \mathbb{R}^d, \lambda_{\text{attn}}, \lambda_{\text{mlp}} > 0, p \in (0,1)$ with $\kappa = (1-p)^{-1/2}$
2: **Output:** $x_{\text{out}} \in \mathbb{R}^{T \times d}$
3: $s_x^2 \leftarrow \frac{1}{d} \sum_{j=1}^d x_{:,j}^2$          ▷ Input variance
4: **parallel do**
    **(A) Variance-approximation**
    $\hat{s}_{x'}^2 \leftarrow s_x^2 + \frac{1}{Td} \|W_V W_O\|_F^2 \left(\frac{\lambda_{\text{attn}}}{\kappa}\right)^2$

    **(B) Attention path**
    *# BHyT before attention*
    $s_x \leftarrow \sqrt{s_x^2}$
    $\alpha_{\text{attn}} \leftarrow \frac{\lambda_{\text{attn}}}{\kappa s_x}$
    $z_{\text{attn}} \leftarrow \gamma_{\text{attn}} \odot \tanh(\alpha_{\text{attn}} \odot x)$          ▷ BHyT$_{\text{attn}}$
    $h_{\text{attn}} \leftarrow \mathcal{A}(z_{\text{attn}})$          ▷ Self-attention layer
5: **end parallel**

6: $x' \leftarrow x + h_{\text{attn}}$          ▷ Residual connection
    *# BHyT before MLP*
7: $\hat{s}_{x'} \leftarrow \sqrt{\hat{s}_{x'}^2}$
8: $\alpha_{\text{mlp}} \leftarrow \frac{\lambda_{\text{mlp}}}{\kappa \hat{s}_{x'}}$
9: $z_{\text{mlp}} \leftarrow \gamma_{\text{mlp}} \odot \tanh(\alpha_{\text{mlp}} \odot x')$          ▷ BHyT$_{\text{mlp}}$
10: $h_{\text{mlp}} \leftarrow \text{MLP}(z_{\text{mlp}})$          ▷ MLP layer
11: $x_{\text{out}} \leftarrow x' + h_{\text{mlp}}$
12: **return** $x_{\text{out}}$          ▷ Residual connection

---

effectively reducing the latency overhead associated with statistical calculations. As shown in Figure 2, our approximated variance aligns closely with the actual variance of intermediate hidden states. Algorithm 1 details this parallelized process within the Transformer block.

### 3.3 DEPTH-WISE VARIANCE PROPAGATION AND STABILITY OF BHyT

We consider the standard residual update $x'_\ell = x_\ell + \text{Attn}(f(x_\ell))$, $x_{\ell+1} = x'_\ell + \text{MLP}(f(x'_\ell))$. Under the variance models established in Theorem 3.5 (attention path) and Theorem B.1 in Appendix B.4 (MLP path), the layer-wise variance recursion (see Appendix B.5.2) can be written in terms of normalized variance gains and a scalar amplification function. For each method normalization function $m$, we define

$$u_{\text{Attn},\ell}^{(m)} := \frac{C_{\text{Attn}} s_{f(x_\ell)}^{(m)\,2}}{s_{x_\ell}^{(m)2}}, \quad u_{\text{MLP},\ell}^{(m)} := \frac{C_{\text{MLP}} s_{f(x'_\ell)}^{(m)\,2}}{s_{x'_\ell}^{(m)2}}, \quad g(\rho, u) := 1 + u + 2\rho\sqrt{u}. \tag{4}$$

Here $\rho \in [-1, 1]$ denotes the correlation coefficients associated with the residual connections. Then the variance at depth $L$ can be written as

$$s_{x_L}^{(m)2} = s_{x_1}^{(m)2} \prod_{\ell=1}^{L-1} g(\rho_1, u_{\text{Attn},\ell}^{(m)}) \, g(\rho_2, u_{\text{MLP},\ell}^{(m)}), \tag{5}$$

so comparing LNS and BHyT reduces to comparing $u_{\text{Attn},\ell}^{(m)}, u_{\text{MLP},\ell}^{(m)}$ at each layer.

**Theorem 3.6** (Finite-depth variance bound of BHyT). *Under the assumptions of, and by, Theorems 3.5 and B.1, and using the normalized gains and amplification function $u_{\text{Attn},\ell}^{(m)}, u_{\text{MLP},\ell}^{(m)}, g(\rho, u)$ defined above. Let $x_\ell^{(LNS)}$ and $x_\ell^{(BHyT)}$ be the hidden states of two identical Transformer stacks that differ only in using LNS or BHyT in every block. If the BHyT hyperparameters $(\lambda, \kappa)$ satisfy $\lambda/\kappa < 1/\sqrt[4]{L}$, then for all layers $\ell < L$ the per-layer variance multiplier of BHyT is strictly smaller than that of LNS, and hence*

$$\text{Var}(x_L^{(BHyT)}) < \text{Var}(x_L^{(LNS)}). \tag{6}$$

Theorem 3.6 gives a finite-depth variance bound for BHyT: if $\lambda/\kappa < 1/(\sqrt[4]{L})$, then for every layer $\ell < L$, the variance multiplier of BHyT is no larger than that of LNS, so the output variance at depth $L$ is smaller. For example, choosing $\lambda = 3$ and $\kappa = 10$ yields $L \approx 123$, up to which BHyT has a smaller output variance than LNS. This suggests that, for typical model depths, BHyT can be more stable than LNS in terms of variance growth.

# 4 EXPERIMENTS & RESULTS

In this section, we present a comprehensive set of experiments designed to evaluate the effectiveness of our proposed method, BHyT. Our evaluation is structured around three primary objectives: 1) **Stability**, assessing BHyT's ability to mitigate the "curse of depth" by analyzing activation growth across layers; 2) **Efficiency**, comparing its computational throughput against existing normalization methods; and 3) **Performance**, measuring its effectiveness in both pretraining and supervised fine-tuning (SFT) contexts to ensure it achieves competitive results.

## 4.1 EXPERIMENTAL SETUP

**Models.** We conduct experiments with two model scales from the Llama-3.2 family: a 1B-parameter model with 16 layers and a 3B-parameter model with 28 layers. Rather than using pretrained checkpoints, we instantiate only the architectures and train all models from scratch. This setup allows us to explicitly examine the effect of increasing model depth on stability and efficiency, and to highlight the scalability benefits of our proposed BHyT method.

**Training data and protocols.** Pretraining is performed on the C4 corpus, a large-scale English text dataset widely used in language model pretraining (Raffel et al., 2020). For supervised fine-tuning (SFT), we employ an instruction-tuning dataset: Lima1k (Zhou et al., 2023), which contains carefully curated, high-quality instructions that provide broader coverage with a larger number of instruction–response pairs. This combination enables us to evaluate both high-quality, low-resource, and large-scale instruction-tuning scenarios.

Following pretraining and SFT, we assess model performance on a diverse suite of seven language understanding and reasoning benchmarks: ARC-e (Clark et al., 2018), PIQA (Bisk et al., 2020), HellaSwag (Zellers et al., 2019), OpenBookQA (Mihaylov et al., 2018), Winogrande (Sakaguchi et al., 2021), MMLU (Hendrycks et al., 2020), and BoolQ (Mousi et al., 2024). These benchmarks collectively evaluate both factual knowledge and reasoning abilities. We report task-specific accuracy and macro-average accuracy across tasks as the primary evaluation metric.

**Frameworks and tooling.** All pretraining and SFT experiments are implemented using the Llama Factory framework (Zheng et al., 2024), which provides efficient fine-tuning utilities and ensures consistent implementation across methods. For downstream evaluation, we use the lm-evaluation-harness toolkit (Gao et al., 2024), adopting its standardized task configurations to ensure comparability with prior work. Accuracy scores are computed directly by the harness without modification, ensuring the reproducibility of reported results.

**Hyperparameter setup** We select hyperparameters from a shared sweep range to ensure a fair comparison across all methods. Each method identifies its optimal configuration by choosing the hyperparameters that yield the lowest evaluation loss after the pretrained C4 datasets reach 20K steps, and we use this configuration for the full training. The detailed hyperparameter search ranges, the final selected configuration for each method, and the full experimental setup—including hardware and parallelism strategy—are provided in the Appendix C.

## 4.2 PRETRAINING LLMS

We compare BHyT against several baselines: RMSNorm (Zhang & Sennrich, 2019), the default normalization used in the Llama family; LNS (Sun et al., 2025) and Peri-LN Kim et al. (2025), both designed to enhance training stability; and DyT (Zhu et al., 2025), which replaces the conventional Pre-LN with a $\tanh(\cdot)$-based function having learnable scalar parameter $\alpha_{\text{DyT}}$. We conducted pretraining on the C4 dataset. The 1B model was optimized with a sequence length of 1024, a batch

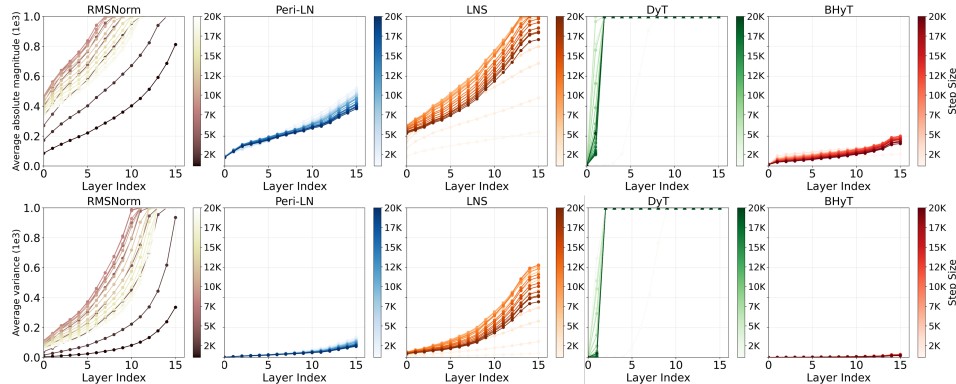

Figure 3: Layer-wise analysis of output statistics on the Llama-1B model. The top row shows the average absolute magnitude of activations, and the bottom row shows the average layer-wise variance. RMSNorm and DyT exhibit rapid growth with depth, reflecting the curse of depth and instability in deeper networks. LNS and Peri-LN suppress this escalation, and BHyT further stabilizes activation variability, yielding lower magnitude and variance than LNS.

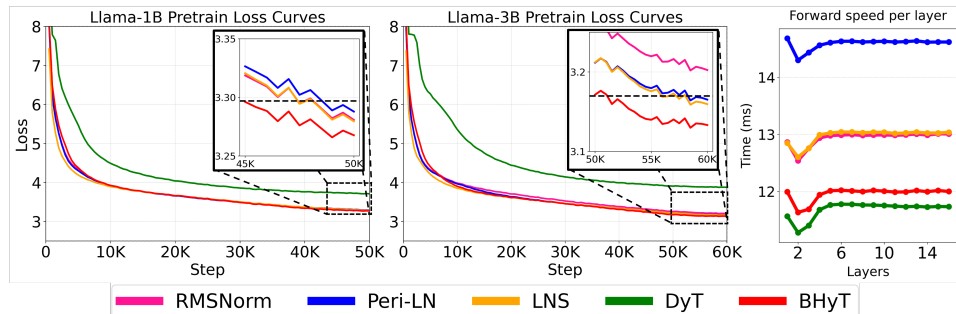

Figure 4: Pretraining performance and forward speed comparison. The first two figures denote loss curves for Llama-1B and Llama-3B, respectively. These show that BHyT achieves stable and competitive convergence, surpassing normalization-based baselines. Notably, compared to normalization-based baselines, BHyT reaches the same loss level fastest towards the end of training. The rightmost figure denotes the Llama-1B forward path speed comparison, highlighting that DyT is the fastest, whereas Peri-LN is the slowest. Overall, BHyT provides the best trade-off, combining reliable convergence with high training efficiency.

size of 32, and for 50,000 steps, amounting to ∼1.64B tokens. The 3B model used the same configuration but was trained for 60,000 steps, resulting in ∼1.97B tokens. To further evaluate each method under a more realistic large-scale setting, we additionally extend the pretraining budget to 20B tokens and conduct experiments on both the 1B and 3B models.

Figure 3 presents the layer-wise evolution of activation magnitude and variance for the Llama-1B model[1]. RMSNorm shows clear instability, with both quantities increasing rapidly with depth. DyT shows a similar trend, suffering from explosive growth in deeper layers; notably, we observed that DyT's training stability is highly sensitive to the learning rate (see Appendix D.2). In contrast, similar to Peri-LN and LNS, BHyT demonstrates a controlled, linear increase in statistics across layers. However, BHyT is distinguished by a noticeably gentler slope compared to these baselines, effectively suppressing depth-wise drift. This behavior leads to more stable optimization while retaining efficiency.

Figure 4 shows the pretraining loss curves for Llama-1B and Llama-3B, alongside the forward pass speed per layer. The results demonstrate that BHyT achieves stable and competitive convergence, notably reaching the target loss level faster than normalization-based baselines towards the end of training. Besides, BHyT significantly outperforms the slower Peri-LN by approximately 17.7% and

---

[1]To ensure a fair comparison, we evaluate all methods under a fixed hyperparameter setting (Learning rate: $3e{-}4$, Warm-up ratio: $1e{-}1$) at 20K training steps.

Table 1: Pretraining (PT)-only evaluation for Llama-1B and Llama-3B; downstream benchmarks in a 5-shot setting, averaged over five training seeds. The results demonstrate that BHyT achieves performance comparable to or superior to strong baselines while maintaining high computational speed. PPL denotes the perplexity score.

| | Llama-1B | | | | | | | | | | |
|---|---|---|---|---|---|---|---|---|---|---|---|
| Method | PT Train Loss | PT Eval Loss | PT Eval PPL | Arc-e | PIQA | Hellaswag | OpenBookQA | Winogrande | MMLU | BoolQ | Avg. |
| RMSNorm | 3.281 | 3.272 | 26.353 | $30.97_{\pm0.20}$ | $62.89_{\pm0.74}$ | $32.77_{\pm0.09}$ | $32.32_{\pm0.94}$ | $\mathbf{50.54}_{\pm0.48}$ | $\mathbf{25.70}_{\pm0.01}$ | $56.26_{\pm0.44}$ | $41.64_{\pm0.10}$ |
| Peri-LN | 3.288 | 3.279 | 26.545 | $\mathbf{31.63}_{\pm0.32}$ | $\mathbf{63.07}_{\pm0.43}$ | $32.05_{\pm0.17}$ | $32.40_{\pm0.80}$ | $49.41_{\pm1.64}$ | $24.91_{\pm0.00}$ | $58.05_{\pm0.34}$ | $41.65_{\pm0.12}$ |
| LNS | 3.280 | 3.271 | 26.342 | $31.39_{\pm0.24}$ | $62.94_{\pm0.64}$ | $\mathbf{32.80}_{\pm0.16}$ | $33.50_{\pm0.58}$ | $50.37_{\pm0.91}$ | $24.88_{\pm0.01}$ | $56.67_{\pm0.37}$ | $41.79_{\pm0.26}$ |
| DyT | 3.709 | 3.696 | 40.294 | $29.28_{\pm0.48}$ | $58.84_{\pm0.58}$ | $26.51_{\pm0.20}$ | $30.85_{\pm0.87}$ | $50.34_{\pm0.68}$ | $24.90_{\pm0.00}$ | $39.08_{\pm0.31}$ | $37.11_{\pm0.11}$ |
| BHyT | 3.268 | 3.254 | 25.908 | $30.50_{\pm0.16}$ | $62.42_{\pm0.41}$ | $32.11_{\pm0.10}$ | $\mathbf{33.88}_{\pm0.73}$ | $50.15_{\pm0.34}$ | $25.19_{\pm0.02}$ | $\mathbf{61.86}_{\pm0.17}$ | $\mathbf{42.30}_{\pm0.13}$ |

| | Llama-3B | | | | | | | | | | |
|---|---|---|---|---|---|---|---|---|---|---|---|
| Method | PT Train Loss | PT Eval Loss | PT Eval PPL | Arc-e | PIQA | Hellaswag | OpenBookQA | Winogrande | MMLU | BoolQ | Avg. |
| RMSNorm | 3.203 | 3.180 | 24.040 | $31.01_{\pm0.21}$ | $\mathbf{66.57}_{\pm0.41}$ | $\mathbf{41.52}_{\pm0.25}$ | $\mathbf{33.92}_{\pm0.33}$ | $50.50_{\pm0.78}$ | $25.67_{\pm0.02}$ | $37.94_{\pm0.12}$ | $41.02_{\pm0.18}$ |
| Peri-LN | 3.165 | 3.142 | 23.156 | $31.82_{\pm0.38}$ | $64.52_{\pm0.24}$ | $36.05_{\pm0.11}$ | $32.28_{\pm0.59}$ | $49.30_{\pm0.52}$ | $25.99_{\pm0.00}$ | $57.46_{\pm0.31}$ | $42.49_{\pm0.25}$ |
| LNS | 3.160 | 3.139 | 23.091 | $\mathbf{31.88}_{\pm0.42}$ | $64.68_{\pm0.46}$ | $36.25_{\pm0.10}$ | $32.45_{\pm0.53}$ | $51.18_{\pm0.75}$ | $\mathbf{26.87}_{\pm0.01}$ | $37.83_{\pm0.08}$ | $40.16_{\pm0.11}$ |
| DyT | 3.877 | 3.855 | 47.244 | $27.69_{\pm0.27}$ | $59.20_{\pm0.15}$ | $25.96_{\pm0.17}$ | $31.85_{\pm0.38}$ | $49.17_{\pm1.28}$ | $25.87_{\pm0.02}$ | $48.05_{\pm0.56}$ | $38.26_{\pm0.12}$ |
| BHyT | 3.133 | 3.107 | 22.346 | $31.84_{\pm0.15}$ | $65.08_{\pm0.39}$ | $36.48_{\pm0.04}$ | $31.76_{\pm0.33}$ | $\mathbf{51.62}_{\pm0.66}$ | $25.70_{\pm0.01}$ | $60.84_{\pm0.29}$ | $\mathbf{43.44}_{\pm0.16}$ |

demonstrates 7.7% faster forward computation than RMSNorm. Consequently, BHyT provides the best trade-off, combining reliable convergence with high training efficiency.

Table 1 shows the pretraining evaluation results, including training loss. The results confirm that BHyT consistently achieves the lowest pretraining loss and superior average accuracy, validating its robust scalability and performance advantage.

## 4.3 INSTRUCTION FINE-TUNING AND BENCHMARK EVALUATION

Table 2: Supervised fine-tuning (SFT) results across five training seeds for Llama-1B and Llama-3B. Reported values include SFT losses and 5-shot downstream benchmark accuracies; BHyT attains lower SFT loss and competitive performance, indicating effective transfer of its pretraining stability to instruction-tuned settings.

| | Llama-1B | | | | | | | | | | |
|---|---|---|---|---|---|---|---|---|---|---|---|
| Method | PT Eval Loss | SFT Train Loss | SFT Eval Loss | Arc-e | PIQA | Hellaswag | OpenBookQA | Winogrande | MMLU | BoolQ | Avg. |
| RMSNorm | 3.272 | $3.200_{\pm0.011}$ | $3.664_{\pm0.144}$ | $26.09_{\pm0.13}$ | $50.58_{\pm0.23}$ | $26.16_{\pm0.07}$ | $30.56_{\pm0.09}$ | $47.18_{\pm0.51}$ | $26.66_{\pm0.13}$ | $37.85_{\pm0.03}$ | $35.01_{\pm0.11}$ |
| Peri-LN | 3.279 | $3.113_{\pm0.012}$ | $3.594_{\pm0.147}$ | $26.68_{\pm0.16}$ | $51.35_{\pm0.10}$ | $25.76_{\pm0.17}$ | $30.80_{\pm0.24}$ | $48.79_{\pm1.02}$ | $\mathbf{26.83}_{\pm0.10}$ | $57.16_{\pm0.85}$ | $38.20_{\pm0.25}$ |
| LNS | 3.271 | $3.118_{\pm0.012}$ | $3.619_{\pm0.145}$ | $27.26_{\pm0.18}$ | $50.08_{\pm0.22}$ | $24.80_{\pm0.04}$ | $\mathbf{34.72}_{\pm0.18}$ | $\mathbf{50.94}_{\pm0.79}$ | $24.78_{\pm0.01}$ | $37.83_{\pm0.00}$ | $35.77_{\pm0.12}$ |
| DyT | 3.696 | $3.226_{\pm0.010}$ | $3.747_{\pm0.135}$ | $30.98_{\pm0.07}$ | $59.53_{\pm0.15}$ | $27.08_{\pm0.13}$ | $27.64_{\pm0.26}$ | $50.56_{\pm0.66}$ | $25.10_{\pm0.08}$ | $60.68_{\pm0.43}$ | $40.22_{\pm0.21}$ |
| BHyT | 3.254 | $\mathbf{2.841}_{\pm0.012}$ | $\mathbf{3.288}_{\pm0.132}$ | $\mathbf{32.49}_{\pm0.22}$ | $\mathbf{64.04}_{\pm0.13}$ | $\mathbf{32.53}_{\pm0.10}$ | $30.76_{\pm0.54}$ | $50.92_{\pm0.45}$ | $24.64_{\pm0.04}$ | $\mathbf{62.01}_{\pm0.05}$ | $\mathbf{42.48}_{\pm0.07}$ |

| | Llama-3B | | | | | | | | | | |
|---|---|---|---|---|---|---|---|---|---|---|---|
| Method | PT Eval Loss | SFT Train Loss | SFT Eval Loss | Arc-e | PIQA | Hellaswag | OpenBookQA | Winogrande | MMLU | BoolQ | Avg. |
| RMSNorm | 3.272 | $2.646_{\pm0.011}$ | $3.217_{\pm0.130}$ | $\mathbf{37.67}_{\pm0.23}$ | $66.96_{\pm0.29}$ | $31.70_{\pm0.12}$ | $31.12_{\pm0.18}$ | $\mathbf{51.22}_{\pm1.60}$ | $25.99_{\pm0.04}$ | $53.31_{\pm2.02}$ | $39.83_{\pm0.21}$ |
| Peri-LN | 3.279 | $\mathbf{2.614}_{\pm0.011}$ | $3.178_{\pm0.132}$ | $36.80_{\pm0.27}$ | $\mathbf{67.05}_{\pm0.12}$ | $\mathbf{37.13}_{\pm0.05}$ | $\mathbf{31.44}_{\pm0.61}$ | $49.06_{\pm0.40}$ | $\mathbf{26.36}_{\pm0.02}$ | $52.75_{\pm1.54}$ | $42.45_{\pm0.29}$ |
| LNS | 3.271 | $2.652_{\pm0.011}$ | $3.157_{\pm0.132}$ | $34.49_{\pm0.17}$ | $65.59_{\pm0.16}$ | $28.96_{\pm0.54}$ | $28.96_{\pm0.54}$ | $50.61_{\pm0.55}$ | $26.33_{\pm0.02}$ | $52.69_{\pm0.66}$ | $41.78_{\pm0.20}$ |
| DyT | 3.855 | $3.361_{\pm0.010}$ | $3.971_{\pm0.132}$ | $29.78_{\pm0.40}$ | $58.42_{\pm0.53}$ | $25.95_{\pm0.17}$ | $28.28_{\pm0.90}$ | $49.53_{\pm0.79}$ | $25.82_{\pm0.13}$ | $\mathbf{56.64}_{\pm1.02}$ | $38.87_{\pm0.17}$ |
| BHyT | 3.254 | $2.693_{\pm0.012}$ | $\mathbf{3.130}_{\pm0.133}$ | $34.61_{\pm0.20}$ | $66.47_{\pm0.27}$ | $36.95_{\pm0.12}$ | $29.68_{\pm0.46}$ | $51.14_{\pm1.02}$ | $26.06_{\pm0.05}$ | $58.21_{\pm0.81}$ | $\mathbf{42.88}_{\pm0.19}$ |

We apply LoRA-based supervised fine-tuning to assess whether the stability achieved during pretraining carries over to instruction-tuned performance. Table 2 summarizes the SFT losses and downstream benchmark accuracies for the Llama-1B and Llama-3B. Across both scales, BHyT maintains the stable optimization observed during pretraining and transfers this advantage effectively to the SFT stage. Notably, BHyT attains lower SFT loss and competitive accuracy on a broad set of benchmarks, demonstrating that its depth-wise stability and controlled activation dynamics continue to support reliable adaptation beyond pretraining.

## 4.4 INFERENCE SPEED AND THROUGHPUT ANALYSIS

To evaluate practical deployment efficiency, we benchmark generation throughput using a Hugging Face-based serving framework. We fixed the input length at 512 tokens and measured generation speeds across varying max new token lengths (128, 512, and 1024) with a maximum sequence length of 4096 for Llama-1B.

Table 3 presents the results, where values represent the mean $\pm$ standard deviation across five trials, and percentages in parentheses denote the relative speed difference compared to BHyT. BHyT demonstrates superior efficiency compared to standard normalization layers. At 128 output tokens, BHyT (53.5 tokens/s) is approximately 5.6% faster than RMSNorm and 12.9% faster than Peri-LN, effectively mitigating the overhead often associated with advanced normalization techniques. Complete results are provided in Appendix D.4

Table 3: Comparison of generation throughput (tokens/s)

| Tokens per second | |
| --- | --- |
| RMSNorm | $50.5_{\pm 1.1}$ $(-5.6\%)$ |
| Peri-LN | $46.6_{\pm 0.7}$ $(-12.9\%)$ |
| LNS | $50.6_{\pm 0.5}$ $(-5.4\%)$ |
| DyT | $55.6_{\pm 0.2}$ $(+3.9\%)$ |
| BHyT | $53.5_{\pm 0.4}$ |

## 4.5 ANALYSIS ON SCALABILITY AND GENERALIZATION

Table 4: Performance of Llama-3B pretrained on 20B tokens, evaluated before and after supervised fine-tuning (SFT). BHyT consistently outperforms the stability-oriented baseline, Peri-LN, achieving lower losses and higher downstream accuracy.

| Method | PT Train Loss | PT Eval Loss | PT Wall time | SFT Tr Loss | SFT Eval Loss | Arc-e | PIQA | Hellaswag | OpenBookQA | Winogrande | MMLU | BoolQ | Avg. |
| --- | --- | --- | --- | --- | --- | --- | --- | --- | --- | --- | --- | --- | --- |
| Llama-3B (Pretrained on 20B tokens only) | | | | | | | | | | | | | |
| Peri-LN | 2.811 | 2.812 | 238h | - | - | $41.03_{\pm 0.56}$ | $68.93_{\pm 0.67}$ | $47.47_{\pm 0.21}$ | $30.64_{\pm 0.99}$ | $51.70_{\pm 0.26}$ | $27.02_{\pm 0.00}$ | $47.19_{\pm 0.74}$ | $44.85_{\pm 0.27}$ |
| BHyT | **2.756** | **2.760** | **171.4h** | - | - | $\mathbf{45.93}_{\pm 0.46}$ $\mathbf{70.09}_{\pm 0.14}$ | $\mathbf{50.97}_{\pm 0.14}$ | $\mathbf{31.32}_{\pm 1.00}$ | $\mathbf{52.12}_{\pm 0.72}$ | $\mathbf{27.62}_{\pm 0.00}$ | $\mathbf{55.69}_{\pm 0.86}$ | $\mathbf{47.68}_{\pm 0.50}$ |
| Llama-3B (Pretrained on 20B tokens & SFT) | | | | | | | | | | | | | |
| Peri-LN | 2.811 | 2.812 | 238h | $2.477_{\pm 0.011}$ | $2.830_{\pm 0.10}$ | $50.19_{\pm 0.34}$ | $70.93_{\pm 0.36}$ | $48.39_{\pm 0.15}$ | $\mathbf{31.00}_{\pm 0.58}$ | $51.98_{\pm 0.64}$ | $26.36_{\pm 0.11}$ | $42.43_{\pm 0.99}$ | $45.90_{\pm 0.21}$ |
| BHyT | **2.756** | **2.760** | **171.4h** | $\mathbf{2.468}_{\pm 0.011}$ | $\mathbf{2.764}_{\pm 0.10}$ | $\mathbf{53.83}_{\pm 0.38}$ | $\mathbf{72.36}_{\pm 0.43}$ | $\mathbf{51.62}_{\pm 0.15}$ | $30.60_{\pm 0.57}$ | $\mathbf{53.07}_{\pm 0.47}$ | $\mathbf{26.82}_{\pm 0.10}$ | $\mathbf{49.89}_{\pm 1.51}$ | $\mathbf{48.31}_{\pm 0.27}$ |

To validate BHyT in a more realistic pretraining setting, we extended training to 20B tokens and compared BHyT against Peri-LN. Table 4 reports results for both pretraining-only and post-SFT stages. BHyT achieves lower pretraining loss and perplexity, indicating that our method effectively suppresses depth-wise drift even with larger budgets. Additional results for the 1B model trained on 20B tokens can be found in Appendix D.5.

## 4.6 ABLATION ON VARIANCE APPROXIMATION

Table 5: Impact of variance approximation on training efficiency and performance. We compare the exact variance calculation baselines (RMSNorm, BHyT*) against their approximation counterparts (RMSNorm-Approx, BHyT). The results indicate that the approximation mechanism consistently boosts training speed (step/s) across architectures with negligible degradation in convergence and downstream accuracy.

| Llama-1B | Variance Approx. | PT Train Loss | PT Eval Loss | PT Eval PPL | Train steps per sec. | Arc-e | PIQA | Hellaswag | OpenBookQA | Winogrande | MMLU | BoolQ | Avg. |
| --- | --- | --- | --- | --- | --- | --- | --- | --- | --- | --- | --- | --- | --- |
| RMSNorm | X | 3.281 | *3.272* | 26.353 | 0.346 | *30.97*$_{\pm 0.20}$ *62.89*$_{\pm 0.74}$ *32.77*$_{\pm 0.09}$ | $32.32_{\pm 0.94}$ | *50.54*$_{\pm 0.48}$ | $\mathbf{25.70}_{\pm 0.01}$ | $56.26_{\pm 0.44}$ | $41.64_{\pm 0.10}$ |
| BHyT* | X | **3.266** | **3.254** | **25.885** | 0.335 | $\mathbf{39.82}_{\pm 0.52}$ $\mathbf{64.86}_{\pm 0.50}$ $\mathbf{35.67}_{\pm 0.15}$ | $27.68_{\pm 0.93}$ | $50.40_{\pm 0.80}$ | *25.19*$_{\pm 0.00}$ | $55.34_{\pm 0.67}$ | $\mathbf{42.71}_{\pm 0.15}$ |
| RMSNorm-Approx | O | *3.293* | *3.284* | 26.672 | *0.381* | $31.15_{\pm 0.44}$ $61.98_{\pm 0.57}$ $31.93_{\pm 0.04}$ | *33.08*$_{\pm 0.04}$ | $\mathbf{50.67}_{\pm 0.94}$ | $25.16_{\pm 0.03}$ | $46.32_{\pm 0.26}$ | $40.04_{\pm 0.09}$ |
| BHyT | O | *3.268* | **3.254** | *25.908* | **0.385** | $30.50_{\pm 0.16}$ $62.42_{\pm 0.41}$ $32.11_{\pm 0.10}$ | $\mathbf{33.88}_{\pm 0.73}$ | $50.15_{\pm 0.32}$ | $25.19_{\pm 0.02}$ | $\mathbf{61.86}_{\pm 0.17}$ | *42.30*$_{\pm 0.13}$ |

We additionally examine the impact of the proposed variance approximation on model performance. Table 5 compares the exact variance calculation methods (RMSNorm, BHyT*) against their approximate counterparts (RMSNorm-Approx, BHyT), following the same setup as Table 1. The results demonstrate that employing variance approximation yields significant computational benefits, improving training speed by approximately 10.1% for RMSNorm and 14.9% for BHyT*. Crucially, this substantial gain in throughput comes with negligible degradation in pretraining loss and downstream accuracy, confirming that our approximation design effectively resolves computational bottlenecks without compromising the model's representational capability.

## 5 CONCLUSION

We proposed Bounded Hyperbolic Tanh (BHyT) to reconcile the trade-off between stability and efficiency in deep LLMs by coupling explicit input bounding with a lightweight, parallelizable variance approximation. Validated by theoretical analysis and extensive experiments, BHyT effectively mitigates the curse of depth and significantly accelerates training throughput compared to standard normalization methods. Future work will focus on validating BHyT's scalability in larger-scale pre-training and its versatility in diverse architectures such as VLMs and encoder-decoders. Additionally, we plan to implement hardware-aware optimizations like custom kernel fusion to verify BHyT's practical robustness in high-throughput serving systems, including vLLM.

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

## A    STATEMENT ON THE USE OF LARGE LANGUAGE MODELS

LLMs were employed solely as a writing assistant to perform limited tasks such as grammar checking and improving readability. The core scientific aspects of this work, including the conception of ideas, development of methodology, theoretical and experimental studies, and drafting of the manuscript, were carried out entirely by the authors without contribution from the LLMs.

## B    PROOFS OF THE THEOREMS

### B.1    PROOF OF PROPOSITION 3.1

*Proof.* Since $|x| \leq |x - \mu_x| + |\mu_x|$, it suffices that

$$\mathbb{P}\left(|x - \mu_x| \leq \frac{\lambda}{|\alpha|} - |\mu_x|\right) \geq p \quad \text{with} \quad \frac{\lambda}{|\alpha|} - |\mu_x| > 0.$$

Chebyshev's inequality gives, for $t > 0$,

$$\mathbb{P}(|x - \mu_x| \leq t\,\sigma_x) \geq 1 - \frac{1}{t^2}.$$

Choose $t = \kappa$ so that $1 - 1/t^2 = p$, and match radii by requiring $t\,\sigma_x = \lambda/|\alpha| - |\mu_x|$; this yields $|\alpha| = \lambda/(\kappa\,\sigma_x + |\mu_x|)$ and hence

$$\mathbb{P}(|x - \mu_x| \leq \kappa\,\sigma_x) \geq p,$$

which implies $\mathbb{P}(|\alpha x| \leq \lambda) \geq p$.    □

### B.2    PROOF OF THEOREM 3.4

*Proof.* Let $x \in \mathbb{R}^d$ be a zero-mean input, i.e., $\mu_x = 0$. Define

$$s(x) := \sqrt{\frac{1}{d}\sum_{k=1}^{d} x_k^2 + \varepsilon} \quad \text{and} \quad \phi(x) := \kappa\sqrt{s_x^2 + \varepsilon} + |\mu_x|.$$

Since $\mu_x = 0$ and $s_x^2 = \frac{1}{d}\sum_{k=1}^{d} x_k^2 = \mathbb{E}[x^2]$, we have

$$\phi(x) = \kappa\sqrt{\mathbb{E}[x^2] + \varepsilon} = \kappa\,s(x).$$

Write $G := \mathrm{diag}(\gamma)$ and

$$P(x) := I - \frac{x\,x^\top}{d\,s(x)^2}.$$

For RMS normalization $\mathrm{RMS}(x) = \gamma \odot x/s(x)$, a direct differentiation yields

$$J_{\mathrm{RMS}}(x) = \frac{1}{s(x)}\,G\,P(x).$$

Define $\mathrm{BHyT}^*(x) = \gamma \odot \tanh\big(\alpha_*(x)\,x\big)$ with

$$\alpha_*(x) := \frac{\lambda}{\phi(x)} = \frac{\lambda}{\kappa}\frac{1}{s(x)}.$$

Using $\partial_j s(x) = x_j/(d\,s(x))$ and thus $\partial_j \alpha_*(x) = -\alpha_*(x)\,x_j/(d\,s(x)^2)$, the chain rule gives, for all $i, j$,

$$\frac{\partial}{\partial x_j}\Big[\gamma_i \tanh\big(\alpha_*(x)\,x_i\big)\Big] = \alpha_*(x)\,\gamma_i\,\mathrm{sech}^2\big(\alpha_*(x)\,x_i\big)\Big(\delta_{ij} - \frac{x_i x_j}{d\,s(x)^2}\Big).$$

Let $D_*(x) := \mathrm{diag}\big(\mathrm{sech}^2(\alpha_*(x)\,x_i)\big)$. Then

$$J_{\mathrm{BHyT}^*}(x) = \alpha_*(x)\,G\,D_*(x)\,P(x) = \frac{\lambda}{\kappa}\,D_*(x)\Big(\frac{1}{s(x)}\,G\,P(x)\Big) = \frac{\lambda}{\kappa}\,D_*(x)\,J_{\mathrm{RMS}}(x).$$

Since $D_*(x)$ is diagonal with entries $\mathrm{sech}^2(\cdot) \in [0, 1]$, we have $\|D_*(x)\|_2 \leq 1$. Taking operator norms in the identity above gives

$$\|J_{\mathrm{BHyT}^*}(x)\|_2 = \left\|\frac{\lambda}{\kappa}\,D_*(x)\,J_{\mathrm{RMS}}(x)\right\|_2 \leq \frac{\lambda}{\kappa}\,\|J_{\mathrm{RMS}}(x)\|_2.$$

□

### B.3 PROOF OF THEOREM 3.5

*Proof.* Let $x \in \mathbb{R}^{T \times d}$ be the input, where $T$ is the sequence length and $d$ is the hidden dimension. We consider the value projection $W_V \in \mathbb{R}^{d \times d_V}$ and the output projection $W_O \in \mathbb{R}^{d_V \times d}$. By assumption, the entries of $x$, $W_V$, and $W_O$ are i.i.d. samples from a zero-mean normal distribution.

**Step 1. Uniform attention weights.** For large $T$, the attention weights are approximately uniform:

$$A \approx \frac{1}{T} \mathbf{1}\mathbf{1}^\top \in \mathbb{R}^{T \times T}.$$

Thus, the attention output after the value and output projections becomes

$$h_{\text{attn}} = (A\, x\, W_V)\, W_O \approx \frac{1}{T} \mathbf{1}^\top x\, W_V W_O \in \mathbb{R}^{1 \times d}.$$

**Step 2. Coordinate expansion.** The $k$-th coordinate of $h_{\text{attn}}$ can be written as

$$(h_{\text{attn}})_k = \frac{1}{T} \sum_{t=1}^{T} \sum_{m=1}^{d} x_{t,m}\, (W_V W_O)_{m,k}. \tag{7}$$

**Step 3. Variance of one coordinate.** Conditioning on $W_V$ and $W_O$ and using that the input $x_{t,m}$ is i.i.d zero-mean with per-coordinate variance $\sigma_x^2$, independent of $W_V, W_O$ and $\text{Var}(x_{t,m}) \approx (\frac{\lambda}{\kappa})^2$, we obtain

$$\text{Var}\big((h_{\text{attn}})_k\big) = \frac{1}{T^2} \sum_{t=1}^{T} \sum_{m=1}^{d} (W_V W_O)_{m,k}^2 \cdot \frac{\lambda^2}{\kappa^2}.$$

Simplifying the sum over $t$ gives

$$\text{Var}\big((h_{\text{attn}})_k\big) = \frac{1}{T} \left\| (W_V W_O)_{:,k} \right\|_2^2 \cdot \frac{\lambda^2}{\kappa^2}. \tag{8}$$

**Step 4. Averaging over all coordinates.** Define the mean variance of the attention output as

$$\tilde{s}_{h_{\text{attn}}}^2 = \frac{1}{d} \sum_{k=1}^{d} \text{Var}\big((h_{\text{attn}})_k\big).$$

This choice is natural since the attention output is a $d$-dimensional random vector, and we require a scalar measure of variance. A standard way is to consider the per-dimension variance, i.e., the average variance across coordinates:

$$\tilde{s}_{h_{\text{attn}}}^2 = \frac{1}{d} \mathbb{E}\big[\|h_{\text{attn}}\|_2^2\big].$$

Under the assumptions, $\mathbb{E}[(h_{\text{attn}})_k] = 0$, so $\mathbb{E}[(h_{\text{attn}})_k^2] = \text{Var}((h_{\text{attn}})_k)$, which justifies the above definition.

Substituting Equation 8, we obtain

$$\tilde{s}_{h_{\text{attn}}}^2 = \frac{1}{Td} \left( \sum_{k=1}^{d} \sum_{m=1}^{d} (W_V W_O)_{m,k}^2 \right) \frac{\lambda^2}{\kappa^2}.$$

The double sum is exactly the squared Frobenius norm:

$$\sum_{k=1}^{d} \sum_{m=1}^{d} (W_V W_O)_{m,k}^2 = \|W_V W_O\|_F^2.$$

Therefore, the variance approximation of the attention output is

$$\tilde{s}_{h_{\text{attn}}}^2 \approx \frac{1}{Td} \|W_V W_O\|_F^2 \cdot \frac{\lambda^2}{\kappa^2}$$

$$\square$$

## B.4 Approximation of variance of MLP layer output

**Theorem B.1** (Variance approximation of MLP layer output). *Let $x'_\ell \in \mathbb{R}^{T \times d}$ be the input to the MLP sublayer at layer $\ell$, and let $z^F_\ell = f_\ell(x'_\ell) \in \mathbb{R}^{T \times d}$ be the output of the preceding Pre-LN. Let $\tilde{s}^2_{h_{\mathrm{MLP}},\ell}$ denote the average per-entry variance of the two-layer MLP output at layer $\ell$, and let $\mathrm{Var}(z^F_\ell)$ denote the common element-wise variance of $z^F_\ell$. Under the same assumptions as Theorem 3.5, applied to the MLP path, we have the approximation*

$$\tilde{s}^2_{h_{\mathrm{MLP}},\ell} \approx \frac{1}{d}\|W_{\ell,1}W_{\ell,2}\|^2_F \, \mathrm{Var}(z^F_\ell) = C_{\mathrm{MLP},\ell} \, \mathrm{Var}(z^F_\ell), \qquad (9)$$

*where $C_{\mathrm{MLP},\ell} := \frac{1}{d}\|W_{\ell,1}W_{\ell,2}\|^2_F$ depends only on the MLP weights at layer $\ell$ and is independent of the specific choice of $f_\ell$.*

*Proof.* We track the variance for a single token and then average over tokens. For a fixed layer $\ell$, write $z^F_\ell = f_\ell(x'_\ell) \in \mathbb{R}^{T \times d}$ and let $z_i \in \mathbb{R}^d$ be the $i$-th row. By the same symmetry assumption as in Theorem 3.5, all entries satisfy $\mathrm{Var}((z_i)_m) = \mathrm{Var}(z^F_\ell)$ and are weakly correlated across $m$.

The position-wise MLP at layer $\ell$ is

$$\mathrm{FFN}_\ell(z) = g(zW_{\ell,1})W_{\ell,2},$$

with activation function $g(\cdot)$ acting coordinate-wise. For a single token $z_i$, we write $a_i = z_iW_{\ell,1}$, $b_i = g(a_i)$, and $h_i = b_iW_{\ell,2}$. Using the same variance approximation as in Appendix B.3, the first linear layer gives

$$\mathrm{Var}((a_i)_j) \approx \mathrm{Var}(z^F_\ell) \sum_{m=1}^{d} (W_{\ell,1})^2_{mj}.$$

By the small-variance assumption, the activation function $g(\cdot)$ can be linearized near the origin and approximately preserves variance, so $\mathrm{Var}((b_i)_j) \approx \mathrm{Var}((a_i)_j)$. Applying the same argument to the second linear layer,

$$(h_i)_k = \sum_{j=1}^{d_{\mathrm{FFN}}} (b_i)_j(W_{\ell,2})_{jk},$$

and neglecting covariances of $(b_i)_j$ as in Theorem 3.5, we obtain

$$\mathrm{Var}((h_i)_k) \approx \mathrm{Var}(z^F_\ell) \sum_{j=1}^{d_{\mathrm{FFN}}} (W_{\ell,2})^2_{jk} \sum_{m=1}^{d} (W_{\ell,1})^2_{mj}.$$

At this point, we again approximate the variance amplification of the two-layer MLP by the Frobenius norm of the composed weight matrix, in the same spirit as Theorem 3.5. Introducing $W_\ell := W_{\ell,1}W_{\ell,2} \in \mathbb{R}^{d \times d}$, we approximate the above expression by

$$\mathrm{Var}((h_i)_k) \approx \mathrm{Var}(z^F_\ell) \sum_{m=1}^{d} (W_\ell)^2_{mk}.$$

To obtain the scalar variance of the vector (consistent with the variance statistic in layer normalization), we compute the mean of these element-wise variances across the hidden dimension $d$. Observing that the sum of squared weights corresponds to the squared Frobenius norm, we have:

$$\tilde{s}^2_{h_{\mathrm{MLP}},\ell} = \frac{1}{d}\sum_{k=1}^{d} \mathrm{Var}((h_i)_k) \approx \frac{1}{d}\|W_{\ell,1}W_{\ell,2}\|^2_F \, \mathrm{Var}(z^F_\ell).$$

$\square$

## B.5 Proof of Theorem 3.6

### B.5.1 Variance bound for BHyT

**Lemma B.2** (Variance bound for $\tanh$ in BHyT). *Let $x$ be a scalar random variable with mean $\mu_x = 0$ and variance $\mathrm{Var}(x) = s_x^2 < \infty$, and define*

$$z = \alpha x, \qquad \alpha = \frac{\lambda}{\kappa s_x}, \tag{10}$$

*for some $\lambda > 0$ and $\kappa > 0$. Assume that $|z| \leq \lambda$ almost surely.[2] Then*

$$\left(\frac{\tanh(\lambda)}{\lambda}\right)^2 \mathrm{Var}(z) \;\leq\; \mathrm{Var}\big(\tanh(z)\big) \;\leq\; \mathrm{Var}(z). \tag{11}$$

*In particular, since $\mathrm{Var}(z) = \alpha^2 s_x^2 = \lambda^2/\kappa^2$, we have*

$$\left(\frac{\tanh(\lambda)}{\lambda}\right)^2 \frac{\lambda^2}{\kappa^2} \;\leq\; \mathrm{Var}\big(\tanh(\alpha x)\big) \;\leq\; \frac{\lambda^2}{\kappa^2}. \tag{12}$$

*Proof.* Fix $\lambda > 0$ and consider the function $g(t) = \tanh(t)$ on $[0, \lambda]$. Using convexity of $g$ on this interval, we can write any $t \in [0, \lambda]$ as a convex combination

$$t = (1 - r) \cdot 0 + r \cdot \lambda, \qquad r = \frac{t}{\lambda} \in [0, 1], \tag{13}$$

and obtain

$$g(t) = g\big((1 - r) \cdot 0 + r \cdot \lambda\big) \;\geq\; (1 - r)g(0) + rg(\lambda) = r \tanh(\lambda) = \frac{t}{\lambda} \tanh(\lambda). \tag{14}$$

On the other hand, $g(t) \leq t$ on $[0, \lambda]$ since $\tanh(t) \leq t$ for $t \geq 0$. Therefore, for all $t \in [0, \lambda]$,

$$\frac{\tanh(\lambda)}{\lambda} t \;\leq\; \tanh(t) \;\leq\; t. \tag{15}$$

By odd symmetry of $\tanh$, we obtain the corresponding inequality for all $t \in [-\lambda, \lambda]$:

$$\frac{\tanh(\lambda)}{\lambda} |t| \;\leq\; |\tanh(t)| \;\leq\; |t|. \tag{16}$$

Applying this to the random variable $z$ and using the assumption $|z| \leq \lambda$ almost surely yields

$$\left(\frac{\tanh(\lambda)}{\lambda}\right)^2 z^2 \;\leq\; \tanh^2(z) \;\leq\; z^2 \quad \text{almost surely.} \tag{17}$$

Taking expectations on both sides gives

$$\left(\frac{\tanh(\lambda)}{\lambda}\right)^2 \mathbb{E}[z^2] \;\leq\; \mathbb{E}[\tanh^2(z)] \;\leq\; \mathbb{E}[z^2]. \tag{18}$$

Because $\mu_x = 0$ and $z = \alpha x$ is a linear transform, we have $\mathbb{E}[z] = 0$ and hence $\mathrm{Var}(z) = \mathbb{E}[z^2]$. Moreover $\mathbb{E}[\tanh(z)] = 0$ by odd symmetry and zero-mean $z$, so $\mathrm{Var}(\tanh(z)) = \mathbb{E}[\tanh^2(z)]$. Thus

$$\left(\frac{\tanh(\lambda)}{\lambda}\right)^2 \mathrm{Var}(z) \;\leq\; \mathrm{Var}\big(\tanh(z)\big) \;\leq\; \mathrm{Var}(z). \tag{19}$$

Finally, with $\alpha = \lambda/(\kappa s_x)$ and $\mathrm{Var}(x) = s_x^2$, we have

$$\mathrm{Var}(z) = \alpha^2 \mathrm{Var}(x) = \left(\frac{\lambda}{\kappa s_x}\right)^2 s_x^2 = \frac{\lambda^2}{\kappa^2}, \tag{20}$$

and substituting this into the inequality above gives Equation 12. $\qquad\square$

---

[2]In the main text, this assumption is justified by Proposition 3.1, which shows that $|z| \leq \lambda$ holds with probability at least $p$ when $\alpha$ is chosen as in BHyT. For simplicity we present the argument under the idealized case $p = 1$.

### B.5.2 VARIANCE RECURSION WITH RESIDUAL CONNECTIONS

We next derive the variance recursion for a generic residual block of the form

$$x'_\ell = x_\ell + h_\ell, \qquad x_{\ell+1} = x'_\ell + g_\ell, \tag{21}$$

where $h_\ell = \text{Attn}(f(x_\ell))$ and $g_\ell = \text{MLP}(f(x'_\ell))$. Here, $f(\cdot)$ denotes a normalization layer (e.g., RMSNorm, BHyT, etc.).

**Lemma B.3** (Generic variance recursion)**.** *Under the assumptions and results of Theorems 3.5 and B.1, the variances $s_\ell^2 = Var(x_\ell)$ satisfy*

$$s_\ell'^2 = s_\ell^2 + C_{Attn}\, Var\big(f(x_\ell)\big) + 2\rho_1\, s_\ell\sqrt{C_{Attn}\, Var\big(f(x_\ell)\big)}, \tag{22}$$

*and*

$$s_{\ell+1}^2 = s_\ell'^2 + C_{MLP}\, Var\big(f(x'_\ell)\big) + 2\rho_2\, s'_\ell\sqrt{C_{MLP}\, Var\big(f(x'_\ell)\big)}. \tag{23}$$

*Proof.* By definition,

$$s_\ell'^2 = \text{Var}(x'_\ell) = \text{Var}(x_\ell + h_\ell) = \text{Var}(x_\ell) + \text{Var}(h_\ell) + 2\,\text{Cov}(x_\ell, h_\ell). \tag{24}$$

Theorems 3.5 and B.1 give $\text{Var}(h_\ell) = C_{\text{Attn}}\text{Var}(f(x_\ell))$ and $\text{Cov}(x_\ell, h_\ell) = \rho_1 s_\ell s_{\text{Attn},\ell}$ with $s_{\text{Attn},\ell}^2 = \text{Var}(h_\ell) = C_{\text{Attn}}\text{Var}(f(x_\ell))$. Substituting these relations yields Equation 22. The expression for $s_{\ell+1}^2$ follows analogously from

$$s_{\ell+1}^2 = \text{Var}(x_{\ell+1}) = \text{Var}(x'_\ell + g_\ell) = \text{Var}(x'_\ell) + \text{Var}(g_\ell) + 2\,\text{Cov}(x'_\ell, g_\ell), \tag{25}$$

combined with $\text{Var}(g_\ell) = C_{\text{MLP}}\text{Var}(f(x'_\ell))$ and $\text{Cov}(x'_\ell, g_\ell) = \rho_2 s'_\ell s_{\text{MLP},\ell}$, where $s_{\text{MLP},\ell}^2 = \text{Var}(g_\ell) = C_{\text{MLP}}\text{Var}(f(x'_\ell))$. $\qquad\square$

### B.5.3 SPECIALIZATION TO RMSNORM, LNS AND BHYT

In this section we instantiate the generic variance recursion from the assumptions and results of Theorems 3.5 and B.1, for three concrete choices of $f(\cdot)$: RMSNorm, LNS (Sun et al., 2025) and BHyT. We follow the derivation in Appendix B.5.2.

**Notation.** We denote by

$$s_{x_\ell}^2 = \text{Var}(x_\ell), \quad s_{x'_\ell}^2 = \text{Var}(x'_\ell), \quad s_{f(x_\ell)}^2 = \text{Var}(f(x_\ell)), \quad s_{f(x'_\ell)}^2 = \text{Var}(f(x'_\ell)). \tag{26}$$

The output variances of the attention and MLP sublayers are $s_{\text{Attn},\ell}^2 = \text{Var}(\text{Attn}(f(x_\ell)))$ and $s_{\text{MLP},\ell}^2 = \text{Var}(\text{MLP}(f(x'_\ell)))$, respectively.

### B.5.4 VARIANCE RECURSION AND PRODUCT FORM

The residual block $x'_\ell = x_\ell + \text{Attn}(f(x_\ell))$, $x_{\ell+1} = x'_\ell + \text{MLP}(f(x'_\ell))$ satisfies the scalar variance recursions

$$\text{Var}(x'_\ell) = \text{Var}(x_\ell) + \text{Var}(\text{Attn}(f(x_\ell))) + 2\rho_1 s_{x_\ell} s_{\text{Attn},\ell}, \tag{27}$$

$$\text{Var}(x_{\ell+1}) = \text{Var}(x'_\ell) + \text{Var}(\text{MLP}(f(x'_\ell))) + 2\rho_2 s_{x'_\ell} s_{\text{MLP},\ell}, \tag{28}$$

where $\rho_1, \rho_2$ are covariance coefficients. Using the proportionality $\text{Var}(\text{Attn}(f(x_\ell))) \approx C_{\text{Attn}}\text{Var}(f(x_\ell))$ and $\text{Var}(\text{MLP}(f(x'_\ell))) \approx C_{\text{MLP}}\text{Var}(f(x'_\ell))$, we obtain

$$\text{Var}(x'_\ell) = \text{Var}(x_\ell) + C_{\text{Attn}}\text{Var}(f(x_\ell)) + 2\rho_1 s_{x_\ell} s_{\text{Attn},\ell}, \tag{29}$$

$$\text{Var}(x_{\ell+1}) = \text{Var}(x'_\ell) + C_{\text{MLP}}\text{Var}(f(x'_\ell)) + 2\rho_2 s_{x'_\ell} s_{\text{MLP},\ell}. \tag{30}$$

Dividing the first identity by $s_{x_\ell}^2$ and the second by $s_{x'_\ell}^2$ yields the layer-wise amplification factors

$$\frac{s_{x'_\ell}^2}{s_{x_\ell}^2} = 1 + \frac{C_{\text{Attn}}\, s_{f(x_\ell)}^2}{s_{x_\ell}^2} + 2\rho_1 \sqrt{\frac{C_{\text{Attn}}\, s_{f(x_\ell)}^2}{s_{x_\ell}^2}}, \tag{31}$$

$$\frac{s_{x_{\ell+1}}^2}{s_{x'_\ell}^2} = 1 + \frac{C_{\text{MLP}}\, s_{f(x'_\ell)}^2}{s_{x'_\ell}^2} + 2\rho_2 \sqrt{\frac{C_{\text{MLP}}\, s_{f(x'_\ell)}^2}{s_{x'_\ell}^2}}. \tag{32}$$

For later convenience, define the normalized variance gains

$$u_{\text{Attn},\ell} := \frac{C_{\text{Attn}} \, s_{f(x_\ell)}^2}{s_{x_\ell}^2}, \qquad u_{\text{MLP},\ell} := \frac{C_{\text{MLP}} \, s_{f(x_\ell')}^2}{s_{x_\ell'}^2}, \tag{33}$$

and the scalar amplification function

$$g(\rho, u) := 1 + u + 2\rho\sqrt{u}. \tag{34}$$

Then Equation 31 - Equation 32 can be written compactly as

$$\frac{s_{x_\ell'}^2}{s_{x_\ell}^2} = g(\rho_1, u_{\text{Attn},\ell}), \qquad \frac{s_{x_{\ell+1}}^2}{s_{x_\ell'}^2} = g(\rho_2, u_{\text{MLP},\ell}), \tag{35}$$

and the one-block amplification factor becomes

$$\frac{s_{x_{\ell+1}}^2}{s_{x_\ell}^2} = g(\rho_1, u_{\text{Attn},\ell}) \, g(\rho_2, u_{\text{MLP},\ell}). \tag{36}$$

Iterating from layer 1 to $L-1$ gives the product-recursive form

$$\frac{s_{x_L}^2}{s_{x_1}^2} = \prod_{\ell=1}^{L-1} g(\rho_1, u_{\text{Attn},\ell}) \, g(\rho_2, u_{\text{MLP},\ell}), \tag{37}$$

or equivalently

$$s_{x_L}^2 = s_{x_1}^2 \prod_{\ell=1}^{L-1} g(\rho_1, u_{\text{Attn},\ell}) \, g(\rho_2, u_{\text{MLP},\ell}). \tag{38}$$

**Lemma B.4** (Depth-wise variance under RMSNorm). *Let $f(x)$ be RMSNorm applied coordinate-wise to a $d$-dimensional vector $x = (x_1, \ldots, x_d)$, and let $s_{x_\ell}^2 = \text{Var}(x_\ell)$ and $s_{x_\ell'}^2 = \text{Var}(x_\ell')$ denote the variances before and after the attention sublayer at layer $\ell$. Define*

$$u_{\text{Attn},\ell}^{RMS} = \frac{C_{\text{Attn}} \, \overline{\gamma^2}}{s_{x_\ell}^2}, \qquad u_{\text{MLP},\ell}^{RMS} = \frac{C_{\text{MLP}} \, \overline{\gamma^2}}{s_{x_\ell'}^2}, \tag{39}$$

*where $\overline{\gamma^2} = \frac{1}{d} \sum_{i=1}^{d} \gamma_i^2$ is the average squared RMSNorm scale, and $g(\rho, u) = 1 + u + 2\rho\sqrt{u}$. Then the depth-wise variance under RMSNorm satisfies*

$$s_{x_L}^2 = s_{x_1}^2 \prod_{\ell=1}^{L-1} g\big(\rho_1, u_{\text{Attn},\ell}^{RMS}\big) \, g\big(\rho_2, u_{\text{MLP},\ell}^{RMS}\big). \tag{40}$$

*Proof.* We first compute the variance of RMSNorm. For a $d$-dimensional vector $x = (x_1, \ldots, x_d)$ with $S = \sum_{i=1}^{d} x_i^2$, RMSNorm is defined coordinate-wise as

$$\text{RMSNorm}(x)_i = \gamma_i \frac{x_i}{\sqrt{\frac{1}{d} \sum_{j=1}^{d} x_j^2 + \varepsilon}} \approx \gamma_i \frac{x_i}{\sqrt{S/d}}, \tag{41}$$

where $\gamma_i$ are learnable scales and we set $\varepsilon = 0$ for simplicity.[3] Then

$$\text{RMSNorm}(x)_i^2 = \gamma_i^2 \frac{x_i^2}{S/d} = \gamma_i^2 \, d \, \frac{x_i^2}{S}. \tag{42}$$

Taking expectation and using symmetry across coordinates gives

$$\mathbb{E}[\text{RMSNorm}(x)_i^2] = \gamma_i^2 \mathbb{E}\left[d \, \frac{x_i^2}{S}\right] = \gamma_i^2 \, d \cdot \frac{1}{d} = \gamma_i^2, \tag{43}$$

---

[3]This approximation is standard and is used only to obtain a simple closed-form variance.

so the per-coordinate variance is $\text{Var}(\text{RMSNorm}(x)_i) = \gamma_i^2$ and the average variance across dimensions is

$$s_{\text{RMSNorm}(x)}^2 = \frac{1}{d}\sum_{i=1}^{d}\gamma_i^2 =: \overline{\gamma^2}. \tag{44}$$

Therefore, at layer $\ell$ we have

$$s_{f(x_\ell)}^2 = \overline{\gamma^2}, \qquad s_{f(x_\ell')}^2 = \overline{\gamma^2}, \tag{45}$$

and by definition of the normalized gains (Equation 38)

$$u_{\text{Attn},\ell}^{\text{RMS}} = \frac{C_{\text{Attn}}\, s_{f(x_\ell)}^2}{s_{x_\ell}^2} = \frac{C_{\text{Attn}}\, \overline{\gamma^2}}{s_{x_\ell}^2}, \qquad u_{\text{MLP},\ell}^{\text{RMS}} = \frac{C_{\text{MLP}}\, s_{f(x_\ell')}^2}{s_{x_\ell'}^2} = \frac{C_{\text{MLP}}\, \overline{\gamma^2}}{s_{x_\ell'}^2}. \tag{46}$$

Substituting these expressions into the generic product form

$$s_{x_L}^2 = s_{x_1}^2 \prod_{\ell=1}^{L-1} g\big(\rho_1, u_{\text{Attn},\ell}\big)\, g\big(\rho_2, u_{\text{MLP},\ell}\big), \tag{47}$$

with $u_{\text{Attn},\ell} = u_{\text{Attn},\ell}^{\text{RMS}}$ and $u_{\text{MLP},\ell} = u_{\text{MLP},\ell}^{\text{RMS}}$, we obtain Equation 40. $\qquad\square$

**Lemma B.5** (Depth-wise variance under LNS)**.** *Let $f_{LNS}(x)$ be LNS, i.e. RMSNorm followed by a depth-dependent scaling factor $a_\ell$ at layer $\ell$. Let $s_{x_\ell}^2 = \text{Var}(x_\ell)$ and $s_{x_\ell'}^2 = \text{Var}(x_\ell')$ denote the variances before and after the attention sublayer, and define*

$$u_{\text{Attn},\ell}^{LNS} = \frac{C_{\text{Attn}}\, \overline{\gamma^2}}{\sqrt{\ell}\, s_{x_\ell}^2}, \qquad u_{\text{MLP},\ell}^{LNS} = \frac{C_{\text{MLP}}\, \overline{\gamma^2}}{\sqrt{\ell}\, s_{x_\ell'}^2}, \tag{48}$$

*where $\overline{\gamma^2} = \frac{1}{d}\sum_{i=1}^{d}\gamma_i^2$ is the average squared RMSNorm scale, and $g(\rho, u) = 1 + u + 2\rho\sqrt{u}$ as before. Then the depth-wise variance under LNS satisfies*

$$s_{x_L}^2 = s_{x_1}^2 \prod_{\ell=1}^{L-1} g\big(\rho_1, u_{\text{Attn},\ell}^{LNS}\big)\, g\big(\rho_2, u_{\text{MLP},\ell}^{LNS}\big). \tag{49}$$

*Proof.* By Lemma B.4, RMSNorm alone produces a normalized variance $s_{\text{RMSNorm}(x_\ell)}^2 = \overline{\gamma^2}$ that is independent of the input variance. LNS applies an additional depth-dependent scalar $a_\ell$ on top of RMSNorm, and the curse-of-depth (CoD) analysis of Sun et al. (2025) chooses $a_\ell$ such that the effective variance decays like

$$s_{f_{\text{LNS}}(x_\ell)}^2 \approx a_\ell^2\, \overline{\gamma^2} \approx \frac{\overline{\gamma^2}}{\sqrt{\ell}}. \tag{50}$$

Thus at layer $\ell$ we have

$$s_{f(x_\ell)}^2 = s_{f_{\text{LNS}}(x_\ell)}^2 \approx \frac{\overline{\gamma^2}}{\sqrt{\ell}}, \qquad s_{f(x_\ell')}^2 \approx \frac{\overline{\gamma^2}}{\sqrt{\ell}}, \tag{51}$$

and by the definition of the normalized gains (Equation 38)

$$u_{\text{Attn},\ell}^{\text{LNS}} = \frac{C_{\text{Attn}}\, s_{f(x_\ell)}^2}{s_{x_\ell}^2} \approx \frac{C_{\text{Attn}}\, \overline{\gamma^2}}{\sqrt{\ell}\, s_{x_\ell}^2}, \qquad u_{\text{MLP},\ell}^{\text{LNS}} = \frac{C_{\text{MLP}}\, s_{f(x_\ell')}^2}{s_{x_\ell'}^2} \approx \frac{C_{\text{MLP}}\, \overline{\gamma^2}}{\sqrt{\ell}\, s_{x_\ell'}^2}. \tag{52}$$

Neglecting constant-factor approximations (which can be absorbed into $C_{\text{Attn}}$ and $C_{\text{MLP}}$), we obtain the expressions stated in the lemma. Substituting $u_{\text{Attn},\ell} = u_{\text{Attn},\ell}^{\text{LNS}}$ and $u_{\text{MLP},\ell} = u_{\text{MLP},\ell}^{\text{LNS}}$ into the generic product form

$$s_{x_L}^2 = s_{x_1}^2 \prod_{\ell=1}^{L-1} g\big(\rho_1, u_{\text{Attn},\ell}\big)\, g\big(\rho_2, u_{\text{MLP},\ell}\big) \tag{53}$$

yields Equation 49. $\qquad\square$

**Lemma B.6** (Depth-wise variance bounds under BHyT). *Let $f(x)$ be BHyT applied coordinate-wise, with parameters $(\lambda, \kappa)$ and per-coordinate scale vector $\gamma$, and let $s_{x_\ell}^2 = \mathrm{Var}(x_\ell)$ and $s_{x'_\ell}^2 = \mathrm{Var}(x'_\ell)$ denote the variances before and after the attention sublayer at layer $\ell$. Define the average squared scale $\overline{\gamma^2} = \frac{1}{d} \sum_{i=1}^{d} \gamma_i^2$ and the normalized gains*

$$u_{Attn,\ell}^{BHyT,low} := \frac{C_{Attn} \, \overline{\gamma^2} \left(\frac{\tanh(\lambda)}{\lambda}\right)^2 \frac{\lambda^2}{\kappa^2}}{s_{x_\ell}^2}, \qquad u_{Attn,\ell}^{BHyT,up} := \frac{C_{Attn} \, \overline{\gamma^2} \frac{\lambda^2}{\kappa^2}}{s_{x_\ell}^2}, \tag{54}$$

$$u_{MLP,\ell}^{BHyT,low} := \frac{C_{MLP} \, \overline{\gamma^2} \left(\frac{\tanh(\lambda)}{\lambda}\right)^2 \frac{\lambda^2}{\kappa^2}}{s_{x'_\ell}^2}, \qquad u_{MLP,\ell}^{BHyT,up} := \frac{C_{MLP} \, \overline{\gamma^2} \frac{\lambda^2}{\kappa^2}}{s_{x'_\ell}^2}, \tag{55}$$

*and recall $g(\rho, u) = 1 + u + 2\rho\sqrt{u}$. Then, under the assumptions of, and by, Theorems 3.5 and B.1, the depth-wise variance under BHyT satisfies the two-sided bounds*

$$s_{x_L}^2 \geq s_{x_1}^2 \prod_{\ell=1}^{L-1} g\big(\rho_1, u_{Attn,\ell}^{BHyT,low}\big) \, g\big(\rho_2, u_{MLP,\ell}^{BHyT,low}\big), \tag{56}$$

$$s_{x_L}^2 \leq s_{x_1}^2 \prod_{\ell=1}^{L-1} g\big(\rho_1, u_{Attn,\ell}^{BHyT,up}\big) \, g\big(\rho_2, u_{MLP,\ell}^{BHyT,up}\big). \tag{57}$$

*Proof.* By Lemma B.2, the per-coordinate variance of BHyT satisfies

$$\overline{\gamma^2} \left(\frac{\tanh(\lambda)}{\lambda}\right)^2 \frac{\lambda^2}{\kappa^2} \; \leq \; s_{BHyT(x)}^2 \; \leq \; \overline{\gamma^2} \frac{\lambda^2}{\kappa^2}, \tag{58}$$

where $\overline{\gamma^2}$ is the average of $\gamma_i^2$ across coordinates, and we suppress the layer index for clarity. Applying this bound to the normalization outputs at layer $\ell$ gives

$$\overline{\gamma^2} \left(\frac{\tanh(\lambda)}{\lambda}\right)^2 \frac{\lambda^2}{\kappa^2} \leq s_{f(x_\ell)}^2 \leq \overline{\gamma^2} \frac{\lambda^2}{\kappa^2}, \tag{59}$$

$$\overline{\gamma^2} \left(\frac{\tanh(\lambda)}{\lambda}\right)^2 \frac{\lambda^2}{\kappa^2} \leq s_{f(x'_\ell)}^2 \leq \overline{\gamma^2} \frac{\lambda^2}{\kappa^2}. \tag{60}$$

Substituting these inequalities into the normalized gains

$$u_{Attn,\ell} = \frac{C_{Attn} \, s_{f(x_\ell)}^2}{s_{x_\ell}^2}, \qquad u_{MLP,\ell} = \frac{C_{MLP} \, s_{f(x'_\ell)}^2}{s_{x'_\ell}^2}, \tag{61}$$

we obtain

$$u_{Attn,\ell}^{BHyT,low} \leq u_{Attn,\ell} \leq u_{Attn,\ell}^{BHyT,up}, \qquad u_{MLP,\ell}^{BHyT,low} \leq u_{MLP,\ell} \leq u_{MLP,\ell}^{BHyT,up}. \tag{62}$$

Since $g(\rho, u)$ is increasing in $u$ for $\rho \geq 0$, this implies

$$g\big(\rho_1, u_{Attn,\ell}^{BHyT,low}\big) \leq g\big(\rho_1, u_{Attn,\ell}\big) \leq g\big(\rho_1, u_{Attn,\ell}^{BHyT,up}\big), \tag{63}$$

$$g\big(\rho_2, u_{MLP,\ell}^{BHyT,low}\big) \leq g\big(\rho_2, u_{MLP,\ell}\big) \leq g\big(\rho_2, u_{MLP,\ell}^{BHyT,up}\big). \tag{64}$$

Finally, plugging these into the generic depth-wise product form

$$s_{x_L}^2 = s_{x_1}^2 \prod_{\ell=1}^{L-1} g\big(\rho_1, u_{Attn,\ell}\big) \, g\big(\rho_2, u_{MLP,\ell}\big), \tag{65}$$

yields the lower and upper bounds in Equation 56–Equation 57. $\square$

### B.5.5 PROOF OF THEOREM 3.6

*Proof.* To compare the output variances of BHyT and LNS, we examine their respective layer-wise variance amplification factors derived in Lemma B.5 and Lemma B.6. Since the scalar amplification function $g(\rho, u) = 1 + u + 2\rho\sqrt{u}$ is monotonically increasing with respect to $u$ for $u > 0$ and $\rho \geq 0$, it suffices to compare the normalized variance gain terms $u$ at each layer $l$. From Lemma B.5, the variance gain for LNS at layer $l$ scales as:

$$u_{\text{Attn},l}^{\text{LNS}} \approx \frac{C_{\text{Attn}}\overline{\gamma^2}}{\sqrt{l} \cdot s_{x_l}^2}, \quad u_{\text{MLP},l}^{\text{LNS}} \approx \frac{C_{\text{MLP}}\overline{\gamma^2}}{\sqrt{l} \cdot s_{x_l'}^2}. \tag{66}$$

From Lemma B.2, the upper bound of the variance gain for BHyT at layer $l$ is determined by:

$$u_{\text{Attn},l}^{\text{BHyT, up}} = \frac{C_{\text{Attn}}\overline{\gamma^2}\frac{\lambda^2}{\kappa^2}}{s_{x_l}^2}, \quad u_{\text{MLP},l}^{\text{BHyT, up}} = \frac{C_{\text{MLP}}\overline{\gamma^2}\frac{\lambda^2}{\kappa^2}}{s_{x_l'}^2}. \tag{67}$$

For the output variance of BHyT to be strictly smaller than that of LNS, the amplification factor of BHyT must be smaller than that of LNS at each layer. Comparing the Attention terms (the MLP terms follow analogously), we require:

$$g(\rho_1, u_{\text{Attn},l}^{\text{BHyT, up}}) < g(\rho_1, u_{\text{Attn},l}^{\text{LNS}}). \tag{68}$$

Substituting the explicit forms into the comparison, the inequality holds if the variance scaling factor of BHyT is smaller than the depth-dependent scaling of LNS:

$$\frac{C_{\text{Attn}}\overline{\gamma^2}\frac{\lambda^2}{\kappa^2}}{s_{x_l}^2} < \frac{C_{\text{Attn}}\overline{\gamma^2}}{\sqrt{l} \cdot s_{x_l}^2}. \tag{69}$$

Canceling common terms ($\frac{C_{\text{Attn}}\overline{\gamma^2}}{s_{x_l}^2}$), we obtain the condition for the squared scaling factors:

$$\frac{\lambda^2}{\kappa^2} < \frac{1}{\sqrt{l}}. \tag{70}$$

Taking the square root implies:

$$\frac{\lambda}{\kappa} < \frac{1}{\sqrt[4]{l}}. \tag{71}$$

Since the term $\frac{1}{\sqrt[4]{l}}$ is a decreasing function of $l$, satisfying this condition for the maximum depth $L$ ensures that it holds for all layers $l < L$. Therefore, if the hyperparameters satisfy $\frac{\lambda}{\kappa} < \frac{1}{\sqrt[4]{L}}$, the per-layer variance multiplier of BHyT is strictly smaller than that of LNS for all layers, leading to:

$$\text{Var}(x_L^{(\text{BHyT})}) < \text{Var}(x_L^{(\text{LNS})}). \tag{72}$$

$\square$

## C HYPERPARAMETER SEARCH AND EXPERIMENTAL SETUP

**Hyperparameter Search Protocol** We perform hyperparameter tuning within a shared sweep range to ensure a fair comparison across all normalization methods. For each method, we identify the optimal configuration by selecting the hyperparameters that achieve the lowest evaluation loss after the Llama-1B model completes 20K pretraining steps. This selected configuration is then used for the full pretraining run and subsequently for supervised fine-tuning (SFT).

### C.1 HYPERPARAMETERS FOR PRETRAINING

**Common sweep ranges**

- **Learning rate (LR):** {1e-4, 3e-4, 5e-4, 1e-3, 3e-3}
- **Weight decay (WD):** {0.0, 0.1}
- **Min LR ratio:** {1e-1, 1e-2}
- **Warmup ratio:** {5e-2, 1e-1}

**Method-specific sweep ranges**

- **BHyT:** initial values of $\lambda$: $\{1.0, 2.0, 3.0, 4.0, 5.0\}$
- **DyT:** learnable tanh-scaling parameter initialized using recommended values from the DyT (Zhu et al., 2025): the learnable tanh-scaling parameter is initialized following the values recommended in the original paper (Zhu et al., 2025): for Llama-1B, we set the scalars to $1.0$ (before Attention), $0.5$ (before MLP), and $0.5$ (last layer), while for Llama-3B we use $0.2, 0.05$, and $0.05$, respectively.

## C.2 SFT Hyperparameter Sweep

- **Learning rate (LR):** $\{1e-7, 5e-7, 1e-6, 1e-5, 5e-5, 1e-4\}$
- **Weight decay (WD):** $\{0.0, 0.1\}$
- **Min LR ratio:** $\{1e-1, 1e-2, 1e-3\}$
- **Warmup ratio:** $\{3e-2, 5e-2, 1e-1\}$

## C.3 Final Hyperparameter Configurations

- RMSNorm: $\{$LR: $3e-4$, WD: $0.1$, Min LR ratio: $1e-1$, Warmup ratio: $1e-1\}$
- Peri-LN: $\{$LR: $3e-4$, WD: $0.1$, Min LR ratio: $1e-1$, Warmup ratio: $1e-1\}$
- LNS: $\{$LR: $5e-4$, WD: $0.1$, Min LR ratio: $1e-1$, Warmup ratio: $5e-2\}$
- DyT: $\{$LR: $1e-4$, WD: $0.1$, Min LR ratio: $1e-1$, Warmup ratio: $1e-1\}$
- BHyT: $\{$LR: $5e-4$, WD: $0.1$, Min LR ratio: $1e-2$, Warmup ratio: $1e-1$, $\lambda$ values: $\text{BHyT}_{\text{Attn}}$: $2.0$, $\text{BHyT}_{\text{MLP}}$: $1.0\}$

## C.4 Hardware and system configuration

- **Pretrain and SFT framework:** Llama-Factory (Zheng et al., 2024)
- **Llama-1B GPU:** RTX A6000
- **Llama-3B GPU:** RTX PRO 6000 Blackwell
- **Parallel computation:** For efficient parallel computation, we use FlashAttention-2 (Dao et al., 2022)
- **Distributed training:** DeepSpeed ZeRO-2

# D  Additional experimental results

## D.1 Time to target loss analysis

To assess how efficiently each normalization method reduces loss during pretraining, we measure the time required to reach the loss achieved by BHyT approximately 5,000 steps before its training is completed. For both the Llama-1B and Llama-3B models, we track this target loss and evaluate competing methods at the point when (and if) they reach it. As shown in Table 6, most baselines reach this loss only much later in training or fail to reach it at all within the allotted training budget, indicating substantially slower effective convergence. Peri-LN, selected as the strongest stability-oriented baseline, eventually approaches the target loss but requires notably more wall-clock time due to the additional normalization applied before and after each sublayer. RMSNorm and LNS converge more slowly and often do not reach the target loss during pretraining. DyT is excluded from this comparison. Although DyT achieves higher throughput in training time, it remains highly sensitive to hyperparameter choices and does not consistently reduce pretraining loss, making it unsuitable for a meaningful time-to-loss comparison. Overall, BHyT reaches the target loss significantly earlier than all baselines while maintaining competitive downstream accuracy, demonstrating that BHyT provides both stable and efficient optimization at scale.

Table 6: Time-to-target-loss comparison for Llama-1B and Llama-3B. Each method is evaluated at the loss value that BHyT reaches roughly 5K steps before completing pretraining. Most baselines reach this loss only much later or fail to reach it within the training budget.

| | | | | Llama-1B | | | | | | | |
|---|---|---|---|---|---|---|---|---|---|---|---|
| Method | PT Train Loss | Wall time (hour) | Checkpoint Step | Arc-e | PIQA | Hellaswag | OpenBookQA | Winogrande | MMLU | BoolQ | Avg. |
| RMSNorm | 3.282 | 39.35 | 49K | $31.28_{\pm0.32}$ | $62.52_{\pm0.50}$ | $32.51_{\pm0.19}$ | $32.64_{\pm0.92}$ | $50.47_{\pm0.66}$ | $23.64_{\pm0.00}$ | $50.94_{\pm0.50}$ | $40.57_{\pm0.22}$ |
| Peri-LN | 3.288 | 42.62 | 50K | $\mathbf{31.63}_{\pm0.32}$ | $\mathbf{63.07}_{\pm0.43}$ | $32.05_{\pm0.17}$ | $32.40_{\pm0.80}$ | $49.41_{\pm1.64}$ | $23.07_{\pm0.00}$ | $58.05_{\pm0.34}$ | $41.38_{\pm0.12}$ |
| LNS | 3.281 | 37.65 | 49K | $31.02_{\pm0.41}$ | $62.50_{\pm0.48}$ | $\mathbf{32.82}_{\pm0.13}$ | $\mathbf{32.65}_{\pm0.70}$ | $50.02_{\pm1.19}$ | $23.27_{\pm0.00}$ | $46.15_{\pm0.61}$ | $39.78_{\pm0.35}$ |
| BHyT | 3.289 | $\mathbf{33.22}$ | $\mathbf{46K}$ | $30.87_{\pm0.19}$ | $62.75_{\pm0.33}$ | $32.36_{\pm0.15}$ | $32.48_{\pm0.18}$ | $\mathbf{51.70}_{\pm1.31}$ | $\mathbf{23.94}_{\pm0.00}$ | $\mathbf{59.23}_{\pm0.63}$ | $\mathbf{41.90}_{\pm0.15}$ |
| | | | | Llama-3B | | | | | | | |
| Method | PT Train Loss | Wall time (hour) | Checkpoint Step | Arc-e | PIQA | Hellaswag | OpenBookQA | Winogrande | MMLU | BoolQ | Avg. |
| RMSNorm | 3.203 | 76.55 | 60K | $31.01_{\pm0.21}$ | $\mathbf{66.57}_{\pm0.41}$ | $\mathbf{41.52}_{\pm0.25}$ | $\mathbf{33.92}_{\pm0.33}$ | $50.50_{\pm0.78}$ | $22.98_{\pm0.00}$ | $37.94_{\pm0.12}$ | $40.63_{\pm0.18}$ |
| Peri-LN | 3.165 | 88.72 | 60K | $31.82_{\pm0.38}$ | $64.52_{\pm0.24}$ | $36.05_{\pm0.11}$ | $32.28_{\pm0.59}$ | $49.30_{\pm0.52}$ | $22.99_{\pm0.00}$ | $\mathbf{57.46}_{\pm0.31}$ | $42.06_{\pm0.25}$ |
| LNS | 3.160 | 78.55 | 60K | $31.88_{\pm0.42}$ | $64.68_{\pm0.46}$ | $36.25_{\pm0.10}$ | $32.45_{\pm0.53}$ | $51.18_{\pm0.75}$ | $23.59_{\pm0.00}$ | $37.83_{\pm0.08}$ | $39.69_{\pm0.11}$ |
| BHyT | 3.135 | $\mathbf{64.30}$ | $\mathbf{56K}$ | $\mathbf{32.44}_{\pm0.17}$ | $64.83_{\pm0.18}$ | $36.31_{\pm0.09}$ | $31.96_{\pm0.36}$ | $\mathbf{51.27}_{\pm1.51}$ | $23.39_{\pm0.00}$ | $56.06_{\pm0.71}$ | $\mathbf{42.32}_{\pm0.31}$ |

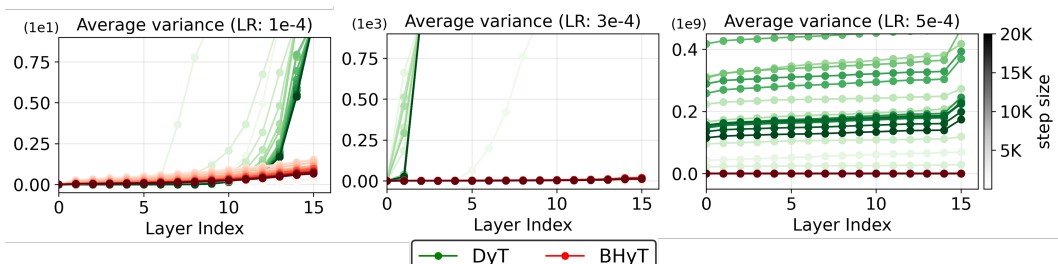

(a) Average per-layer variance in Llama-1B for DyT and BHyT under different learning rates.

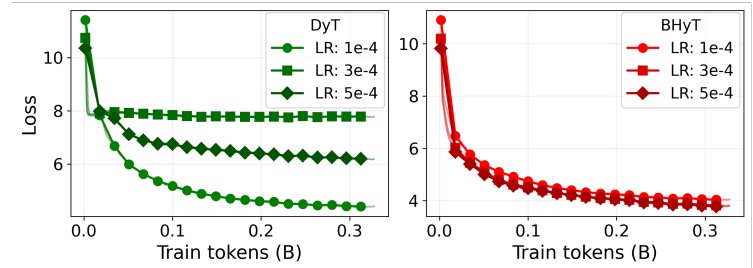

(b) Pretraining loss curves of DyT and BHyT for different learning rates.

Figure 5: Learning-rate robustness of DyT and BHyT with tanh-based normalization in Llama-1B. (a) Average per-layer variance as a function of depth under different learning rates; DyT exhibits rapidly increasing variance at larger learning rates, while BHyT maintains a much smaller variance that grows roughly linearly with depth. (b) Pretraining loss versus training tokens for DyT (left) and BHyT (right) across learning rates, where BHyT converges stably over a wide range of learning rates.

## D.2 TRAINING STABILITY UNDER VARYING LEARNING RATES: DyT VS. BHyT

We compare the training stability of DyT and BHyT, which replace the normalization layer with a tanh-based layer when the learning rate is varied. Figure 5 shows that, for DyT, increasing the learning rate leads to a sharp growth in variance amplification, making pretraining harder to converge. In contrast, BHyT exhibits a much smaller variance scale that grows roughly linearly with depth, and its pretraining converges stably across a wide range of learning rates. These results indicate that BHyT offers robust training stability for large-scale models such as LLMs, enabling more efficient and reliable hyperparameter sweeps.

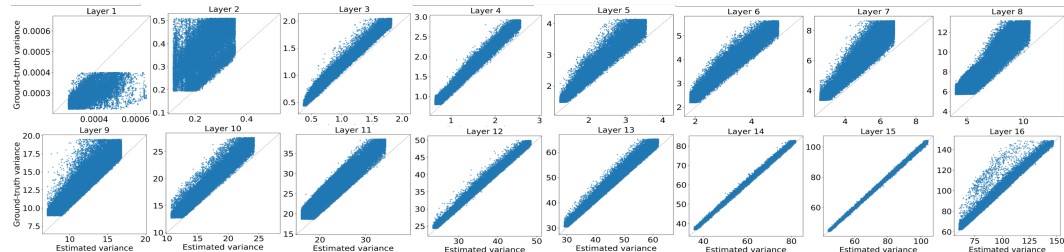

Figure 6: Full layer-wise comparison between the approximated and ground-truth activation variances for the second BHyT layer across all Transformer blocks in Llama-1B. Each subplot corresponds to one layer and shows the estimated variance against the empirical variance computed from randomly sampled C4 inputs.

Table 7: Evaluation of the variance approximation. RMSE quantifies the discrepancy between the approximated and ground-truth variances, while the coefficient of determination $R^2$ measures how well the approximation explains the variability of the ground-truth values. The Pearson correlation coefficient $r$ and the Spearman rank correlation $\rho$ assess linear and rank-based associations, respectively. Higher values of $R^2$, $r$, and $\rho$ indicate better approximation performance.

| | |
|---|---|
| RMSE | $2.124_{\pm 5e-2}$ |
| $R^2$ | $0.99_{\pm 4e-5}$ |
| $r$ | $0.99_{\pm 1e-5}$ |
| $\rho$ | $0.99_{\pm 5e-5}$ |

### D.3 VARIANCE APPROXIMATION

Figure 6 presents a layer-wise comparison between the approximated and ground-truth activation variances for the second BHyT layer across all Transformer blocks in Llama-1B. Table 7 summarizes the quantitative evaluation, showing low RMSE and high $R^2$, Pearson correlation $r$, and Spearman rank correlation $\rho$, which together indicate that the approximation accurately captures both the scale and structure of the true variances.

### D.4 TOKEN GENERATION SPEED

Table 8: Comparison of generation throughput (tokens/s). Values represent mean $\pm$ standard deviation for five-trial, and percentages in parentheses denote the relative speed difference compared to BHyT. The results demonstrate that BHyT achieves competitive inference speeds, consistently outperforming normalization-based baselines.

| Throughput (tokens/s) | Max new token length | | |
|---|---|---|---|
| | 128 | 512 | 1024 |
| RMSNorm | $50.5_{\pm 1.1}$ $(-5.6\%)$ | $38.0_{\pm 0.3}$ $(-3.6\%)$ | $30.1_{\pm 0.2}$ $(-3.5\%)$ |
| Peri-LN | $46.6_{\pm 0.7}$ $(-12.9\%)$ | $34.5_{\pm 0.0}$ $(-12.4\%)$ | $27.0_{\pm 0.1}$ $(-13.5\%)$ |
| LNS | $50.6_{\pm 0.5}$ $(-5.4\%)$ | $37.4_{\pm 0.2}$ $(-5.1\%)$ | $29.5_{\pm 0.1}$ $(-5.4\%)$ |
| DyT | $55.6_{\pm 0.2}$ $(+3.9\%)$ | $40.2_{\pm 1.3}$ $(+2.0\%)$ | $32.3_{\pm 0.2}$ $(+3.5\%)$ |
| BHyT | $53.5_{\pm 0.4}$ | $39.4_{\pm 0.2}$ | $31.2_{\pm 0.2}$ |

Table 8 shows the generation throughput (tokens/s) across different values of the maximum number of new tokens, showing mean $\pm$ standard deviation over five trials and the relative speed difference with respect to BHyT. Across all generation lengths, BHyT consistently achieves higher throughput than normalization-based baselines, indicating that its inference-time advantage is maintained even as the maximum number of generated tokens increases.

## D.5 Pretraining results of Llama-1B on 20B tokens

Table 9: Performance of Llama-1B pretrained on 20B tokens, evaluated before and after supervised fine-tuning (SFT). BHyT consistently outperforms the stability-oriented baseline, Peri-LN, achieving lower losses and higher downstream accuracy.

| Method | PT Train Loss | PT Eval Loss | PT Wall time | SFT Tr Loss | SFT Eval Loss | Arc-e | PIQA | Hellaswag | OpenBookQA | Winogrande | MMLU | BoolQ | Avg. |
|---|---|---|---|---|---|---|---|---|---|---|---|---|---|
| Llama-1B (Pretrained on 20B tokens only) | | | | | | | | | | | | | |
| Peri-LN | 2.870 | 2.846 | 161.7h | - | - | $37.91_{\pm 0.23}$ | $\mathbf{69.16}_{\pm 0.33}$ | $46.77_{\pm 0.10}$ | $\mathbf{31.36}_{\pm 0.82}$ | $\mathbf{50.89}_{\pm 0.96}$ | $\mathbf{27.71}_{\pm 0.00}$ | $51.74_{\pm 0.63}$ | $45.08_{\pm 0.09}$ |
| BHyT | $\mathbf{2.828}$ | $\mathbf{2.802}$ | $\mathbf{135.0h}$ | - | - | $\mathbf{40.98}_{\pm 0.43}$ | $69.13_{\pm 0.34}$ | $\mathbf{49.00}_{\pm 0.07}$ | $30.56_{\pm 0.73}$ | $50.67_{\pm 0.75}$ | $26.83_{\pm 0.00}$ | $\mathbf{55.86}_{\pm 0.38}$ | $\mathbf{46.15}_{\pm 0.23}$ |
| Llama-1B (Pretrained on 20B tokens & SFT) | | | | | | | | | | | | | |
| Peri-LN | 2.870 | 2.846 | 161.7h | $\mathbf{2.522}_{\pm 0.011}$ | $2.856_{\pm 0.107}$ | $47.90_{\pm 0.63}$ | $\mathbf{71.09}_{\pm 0.35}$ | $47.79_{\pm 0.24}$ | $\mathbf{31.24}_{\pm 0.52}$ | $\mathbf{51.98}_{\pm 0.48}$ | $26.68_{\pm 0.01}$ | $48.74_{\pm 0.79}$ | $46.44_{\pm 0.27}$ |
| BHyT | $\mathbf{2.828}$ | $\mathbf{2.802}$ | $\mathbf{135.0h}$ | $2.550_{\pm 0.011}$ | $\mathbf{2.825}_{\pm 0.112}$ | $\mathbf{49.23}_{\pm 0.60}$ | $71.02_{\pm 0.28}$ | $\mathbf{49.42}_{\pm 0.15}$ | $30.72_{\pm 0.84}$ | $51.49_{\pm 0.80}$ | $\mathbf{27.36}_{\pm 0.01}$ | $\mathbf{54.83}_{\pm 0.94}$ | $\mathbf{47.72}_{\pm 0.31}$ |

