# OpenReview forum: "Bounded Hyperbolic Tangent: A Stable and Efficient Alternative to Pre-Layer Normalization in Large Language Models"
_ICLR.cc/2026/Conference — Submitted to ICLR 2026_

### Official Review · Reviewer_fZC6 · 2025-10-30

**Soundness:** 3
**Presentation:** 2
**Contribution:** 3
**Rating:** 6
**Confidence:** 4

**Summary:**

Peri‑LN was proposed to mitigate layer‑wise gradient and variance growth in Transformers but it introduces notable computational overhead. This paper takes an alternative path: it replaces the normalizer with a tanh‑based formulation that explicitly constrains inputs to a preset range, which keeps activations away from saturation and curbs depth‑wise drift. The authors provide theoretical guarantees for stability and show that their formulation does not require the extra normalizing stages used by Peri‑LN while retaining compatibility with standard Transformer blocks.

**Strengths:**

* Conceptually clean formulation that directly bounds activations and targets the underlying cause of depth‑related instability.
* Solid theory that upper‑bounds gradient amplification and supports the intended stability properties.
* Empirical results broadly align with the theory and indicate improved stability of layer statistics across depth.

**Weaknesses:**

* Throughput is reported as higher than Peri‑LN, yet training loss at a fixed number of steps is slightly worse. It remains unclear whether the proposed method would surpass Peri‑LN under equal training compute. Time‑to‑target‑loss or equal‑FLOPs comparisons would make the efficiency claim more convincing.
* The paper should analyze where Peri‑LN’s inefficiency comes from. Is the bottleneck inherent to the algorithmic cost of repeated reductions and memory movement, or chiefly an implementation artifact such as unfused CUDA kernels or suboptimal scheduling? Kernel‑level profiling or theory‑backed complexity analysis would clarify this point.
* The final downstream comparison emphasizes overall averages, but absolute MMLU scores are low. Since a random baseline is 25% on four‑choice tasks, small differences at this scale may be hard to interpret for models trained with limited tokens.
* With the current experimental setup it is difficult to conclude that BHyT is definitively more effective than Peri‑LN. That said, the gains over the RMSNorm baseline are sizable, which suggests BHyT is a strong and practical alternative to Peri‑LN.

**Questions:**

* Not a question but just a minor editing issue: `\citep` should be used in places where citation on points is intended. (e.g., line 094, 095, ...)

---

> ### Author Response · Authors · 2025-11-29
>
> **W1. Time‑to‑target‑loss or equal‑FLOPs comparisons would make the efficiency claim more convincing.**
>
> We thank the reviewer for this insightful comment. We agree that throughput alone is not a sufficient metric for efficiency and that convergence speed relative to wall-clock time (or FLOPs) provides a more realistic comparison.
>
> To address this, **we conducted a Time-to-Target-Loss analysis**, which is detailed in Appendix D.1 of the revised manuscript. **We measured the wall-clock time required for each method to reach the loss value that BHyT achieves approximately 5,000 steps before** its training completion.
>
> The results demonstrate that BHyT reaches the target loss significantly faster than baselines:
>
> Tab. 1. Time-to-target-loss of Llama-1B
> |  Method | PT Train Loss | Wall time (hour) | Checkpoint Step |             Avg. Acc.             |
> |:-------:|:-------------:|:----------------:|:---------------:|:----------------------------:|
> | RMSNorm |     3.282     |       39.35      |       49K       |      $40.57_{\pm{0.22}}$     |
> | Peri-LN |     3.288     |       42.62      |       50K       | $\textit{41.38}_{\pm{0.12}}$ |
> |   LNS   |     3.281     |       37.65      |       49K       |      $39.78_{\pm{0.35}}$     |
> |   BHyT  |     3.289     |       33.22      |       46K       | $\textbf{41.90}_{\pm{0.15}}$ |
>
> As shown above, **while Peri-LN is a strong baseline for stability, it requires notably more wall-clock time to reach the same loss level** due to the computational overhead of applying normalization both before and after sublayers. In contrast, **BHyT achieves comparable convergence much earlier, confirming its superiority in a compute-constrained (equal-time) setting.**
>
> Overall, this time-to-target-loss analysis confirms that **BHyT is more efficient** than Peri-LN under equal-compute/equal-time conditions, not just in terms of raw throughput.
>
> **W2. The paper should analyze where Peri‑LN’s inefficiency comes from.**
>
> Peri-LN's inefficiency is primarily algorithmic and memory-bound. In a standard Transformer block, normalization is typically applied once per sub-layer (Pre-LN or Post-LN). **However, Peri-LN applies normalization both before and after the sub-layers (Attention and FFN).**
>
> This introduces the following overheads:
> - **Increased Memory Access (IO Bound)**: Normalization layers are **memory-bandwidth intensive operations**. **Peri-LN doubles the number of read/write operations for normalization parameters** and activations within each block.
> - **Kernel Launch Overhead**: It requires launching **additional CUDA kernels** **for the extra normalization steps**, which interrupts the computational flow more frequently compared to BHyT or standard RMSNorm.
> - Therefore, the slowdown is an intrinsic limitation of the Peri-LN architecture involving repeated reductions and memory movements, rather than a mere implementation artifact.
> - In practice, **BHyT offers a ~17% faster forward speed** (Fig. 4b in the revised manuscript) and **~12.9% faster token generation** (Tab. 3 in the revised manuscript) **compared to Peri-LN.**

---

> ### Author Response · Authors · 2025-11-29
>
> **W3. The final downstream comparison emphasizes overall averages, but absolute MMLU scores are low**
>
> We appreciate this thoughtful comment and agree that MMLU is a particularly challenging benchmark, especially for relatively small models trained on limited data.
>
> Among our seven benchmarks, MMLU is indeed the hardest task, as it probes broad factual and conceptual knowledge. As a result, it shows lower absolute scores compared to other benchmarks.
>
> To better understand performance on such difficult tasks, during the revision we:
> - **Pretrained Llama-1B and Llama-3B models on 20B tokens of C4**, then applied SFT using the same procedure as in the original submission, and compared Peri-LN and BHyT.
>
> - For Llama-1B, the results after 20B-token pretraining and SFT are:
>
> Tab. 2. Llama-1B (Pretrained on 20B tokens & SFT)
> | Method | PT Train Loss | PT Eval Loss | PT Eval PPL | SFT Tr Loss | SFT Eval Loss |
> |:---:|:---:|:---:|:---:|:---:|:---:|
> | Peri-LN | 2.870 | 2.846 | 17.22 | $\textbf{2.522}_{\pm{0.011}}$ | $2.856_{\pm{0.107}}$ |
> | BHyT | **2.828** | **2.802** | **16.47** | $2.550_{\pm{0.011}}$ | $\textbf{2.825}_{\pm{0.112}}$ |
>
> | Method | Arc-e | PIQA | Hellaswag | OpenBookQA | Winogrande | MMLU | BoolQ | Avg. |
> |:---:|:---:|:---:|:---:|:---:|:---:|:---:|:---:|:---:|
> | Peri-LN | $47.90_{\pm{0.63}}$ | $\textbf{71.09}_{\pm{0.35}}$ | $47.79_{\pm{0.24}}$ | $\textbf{31.24}_{\pm{0.52}}$ | $\textbf{51.98}_{\pm{0.48}}$ | $26.68_{\pm{0.01}}$ | $48.74_{\pm{0.79}}$ | $46.44_{\pm{0.27}}$ |
> | BHyT | $\textbf{49.23}_{\pm{0.60}}$ | $71.02_{\pm{0.28}}$ | $\textbf{49.42}_{\pm{0.15}}$ | $30.72_{\pm{0.84}}$ | $51.49_{\pm{0.80}}$ | $\textbf{27.36}_{\pm{0.01}}$ | $\textbf{54.83}_{\pm{0.94}}$ | $\textbf{47.72}_{\pm{0.31}}$ |
>
> (These results are reported in full as Table 9 in the revised manuscript.)
>
> - Both Peri-LN and BHyT achieve higher MMLU scores when trained on more data.
> - **BHyT yields the best average accuracy and better MMLU than Peri-LN, confirming that its gains are not limited to only easier tasks.**
>
> - Based on this observation, **we expect that as we scale the model size and pretraining tokens further, MMLU scores will continue to improve**, and the **relative advantage of BHyT over Peri-LN will remain or even widen.**
>
> - We are currently training the Llama-3B models under the same 20B-token regime, and we aim to share those results before the discussion period ends.
>
> **W4 With the current experimental setup it is difficult to conclude that BHyT is definitively more effective than Peri‑LN...**
>
> We appreciate this balanced assessment and agree that drawing very strong conclusions requires careful evidence.
>
> We believe **BHyT is a highly competitive method** against both RMSNorm and Peri-LN.
>
> - **Performance**:
>     - Table 1 (pretrain only) and Table 2 (pretrain + SFT) show that **BHyT converges to lower loss and PPL than Peri-LN and achieves better or comparable average downstream performance.**
>     - Extended 20B-token experiments: Table 4 (Llama-1B with 20B pretraining + SFT) shows that **BHyT improves MMLU and average accuracy beyond Peri-LN, even in a more data-rich setting.**
> - **Efficiency**:
>     - **The key advantage of BHyT is speed.** As demonstrated in the **Time-to-Target analysis**, **BHyT reaches target performance 5,000 to 10,000 steps faster than baselines**. Additionally, **BHyT offers a ~17% faster forward speed (Fig. 4b) and ~12.9% faster token generation (Table 3) compared to Peri-LN.**
> - We have softened overly strong language in the paper and now emphasize that **BHyT is competitive with Peri-LN in accuracy while clearly superior in efficiency**, which we believe is well supported by the revised empirical evidence.

---

> ### Author Response · Authors · 2025-12-03
>
> Since the 20B-token pretraining of the LLaMA-3B model was completed before the discussion period, we are sharing the results.
>
> Tab. 3. Llama-3B (Pretrained on 20B tokens only)
> | Method | PT Train Loss | PT Eval Loss | PT Wall time |
> |:---:|:---:|:---:|:---:|
> | Peri-LN | 2.811 | 2.812 | 238h |
> | BHyT | 2.756 | 2.760 | 171.4h |
>
> | Method | Arc-e | PIQA | Hellaswag | OpenBookQA | Winogrande | MMLU | BoolQ | Avg. |
> |:---:|:---:|:---:|:---:|:---:|:---:|:---:|:---:|:---:|
> | Peri-LN | $41.03_{\pm{0.56}}$ | $68.93_{\pm{0.67}}$ | $47.47_{\pm{0.21}}$ | $30.64_{\pm{0.99}}$ | $51.70_{\pm{0.26}}$ | $27.02_{\pm{0.00}}$ | $47.192_{\pm{0.74}}$ | $44.85_{\pm{0.27}}$ |
> | BHyT | $\textbf{45.93}_{\pm{0.46}}$ | $\textbf{70.09}_{\pm{0.14}}$ | $\textbf{50.97}_{\pm{0.14}}$ | $\textbf{31.32}_{\pm{1.00}}$ | $\textbf{52.12}_{\pm{0.72}}$ | $\textbf{27.62}_{\pm{0.00}}$ | $\textbf{55.69}_{\pm{0.86}}$ | $\textbf{47.68}_{\pm{0.5}}$ |
>
> Tab. 4. Llama-3B (Pretrained on 20B tokens & SFT)
>
> | Method | PT Train Loss | PT Eval Loss | PT Wall time | SFT Tr Loss | SFT Eval Loss |
> |:---:|:---:|:---:|:---:|:---:|:---:|
> | Peri-LN | 2.811 | 2.812 | 238h | $2.477_{\pm{0.011}}$ | $2.830_{\pm{0.10}}$ |
> | BHyT | 2.756 | 2.760 | 171.4h | $\textbf{2.468}_{\pm{0.011}}$ | $\textbf{2.764}_{\pm{0.10}}$ |
>
> | Method | Arc-e | PIQA | Hellaswag | OpenBookQA | Winogrande | MMLU | BoolQ | Avg. |
> |:---:|:---:|:---:|:---:|:---:|:---:|:---:|:---:|:---:|
> | Peri-LN | $50.19_{\pm{0.34}}$ | $70.93_{\pm{0.36}}$ | $48.39_{\pm{0.15}}$ | $\textbf{31.00}_{\pm{0.58}}$ | $51.98_{\pm{0.64}}$ | $26.36_{\pm{0.11}}$ | $42.43_{\pm{0.99}}$ | $45.90_{\pm{0.21}}$ |
> | BHyT | $\textbf{53.83}_{\pm{0.38}}$ | $\textbf{72.36}_{\pm{0.43}}$ | $\textbf{51.62}_{\pm{0.15}}$ | $30.60_{\pm{0.57}}$ | $\textbf{53.07}_{\pm{0.47}}$ | $\textbf{26.82}_{\pm{0.10}}$ | $\textbf{49.89}_{\pm{1.51}}$ | $\textbf{48.31}_{\pm{0.27}}$ |
>
> Even with a larger model size and more training tokens, BHyT continued to provide stable and faster training as well as higher evaluation performance compared to Peri-LN.
> These results described in detail in Tab. 4 in Sec. 4.5 of the revised manuscript.

---

### Official Review · Reviewer_5S5H · 2025-10-30

**Soundness:** 3
**Presentation:** 3
**Contribution:** 2
**Rating:** 4
**Confidence:** 4

**Summary:**

The paper proposes Bounded Hyperbolic Tanh (BHyT) as a drop‑in replacement for Pre‑Layer Normalization (typically RMSNorm) in Transformer blocks. The key idea is to pair a tanh nonlinearity with explicit, per‑instance input bounding so activations stay in a non‑saturating regime while avoiding repeated statistic computations.  Empirically, BHyT is evaluated by pretraining 1B/3B Llama‑style models from scratch on C4, then (optionally) SFT on LIMA‑1k and Commonsense‑170K, with downstream evaluation.

**Strengths:**

1 Creative combination of normalization‑free activations with explicit, data‑aware input bounding and a block‑level variance approximation; the latter gives a neat knob to keep overhead small while preserving stability

2 Clear derivations, concise definitions, readable pseudocode,

3 If the approach scales, it could reduce normalization overhead in large LMs without sacrificing training stability

**Weaknesses:**

Please see the questions.

**Questions:**

1 Since the paper introduces  BHyT∗ first, could you please add an ablation directly comparing BHyT∗ and the practical BHyT under the same setup, reporting training loss/validation perplexity, downstream accuracy, and throughput.

2  If the variance can be estimated cheaply, could you apply this estimator to RMSNorm (e.g., replace one normalization with the estimate) and report whether similar speedups and performance are achieved?

3 Please include validation perplexity results of pretraining.

4  Please specify hardware (GPU/TPU model & count), precision, parallelism strategy (DP/TP/PP/ZeRO), global batch size/sequence length, and the measurement protocol.

5  Since LNS is a strong baseline here and LNS reports BoolQ, could you add BoolQ results for parity with the tasks in Table 1?

---

> ### Author Response · Authors · 2025-11-29
>
> **Q1. Comparison of BHyT vs. $\text{BHyT}^*$**
>
> We thank for this helpful suggestion. We have added a direct ablation study comparing BHyT$^*$ (with exact variance) and BHyT (with approximated variance) under the same training setup.
>
> Tab. 1. BHyT vs. $\text{BHyT}^*$
> | Llama-1B | Variance Approx. | PT Train Loss | PT Eval Loss | PT Eval PPL | Train steps per sec. |
> |:---:|:---:|:---:|:---:|:---:|:---:|
> | $\text{BHyT}^{*}$ | X | **3.266** | **3.254** | **25.885** | 0.335 |
> | BHyT | O | 3.268 | **3.254** | 25.908 | **0.385** |
>
> | Llama-1B | Variance Approx. | Arc-e | PIQA | Hellaswag | OpenBookQA | Winogrande | MMLU | BoolQ | Avg. |
> |:---:|:---:|:---:|:---:|:---:|:---:|:---:|:---:|:---:|:---:|
> | $\text{BHyT}^{*}$ | X | $\textbf{39.82}_{\pm{0.52}}$ | $\textbf{64.86}_{\pm{0.50}}$ | $\textbf{35.67}_{\pm{0.15}}$ | $27.68_{\pm{0.93}}$ | $50.40_{\pm{0.80}}$ | $\textbf{24.62}_{\pm{0.00}}$ | $55.34_{\pm{0.67}}$ | $\textbf{42.63}_{\pm{0.15}}$ |
> | BHyT | O | $30.50_{\pm{0.16}}$ | $62.42_{\pm{0.41}}$ | $32.11_{\pm{0.10}}$ | $\textbf{33.88}_{\pm{0.73}}$ | $50.15_{\pm{0.32}}$ | $23.02_{\pm{0.00}}$ | $\textbf{61.86}_{\pm{0.17}}$ | $\textit{41.99}_{\pm{0.13}}$ |
>
> - As expected, BHyT$^*$, which computes the variance exactly at every step, **achieves slightly better training loss** and **benchmark averages** and can be seen as the “oracle” variant in terms of stability.
> - At the same time, The **performance gap between BHyT and BHyT$^*$ is modest**, while **BHyT is significantly faster in terms of training throughput** (0.385 vs. 0.335 steps/sec).
> - This ablation confirms that: **Using the variance approximation does not substantially hurt performance**, and **BHyT achieves a better overall trade-off between stability and efficiency**.
>
> We have added this ablation as Tab. 5 in the revised manuscript.
>
> **Q2. Can you apply this estimator to RMSNorm?**
>
> We appreciate this insightful suggestion.
>
> - Conceptually, **we agree that RMSNorm can also benefit from variance approximation** to reduce the cost of computing exact statistics.
> - However, **our primary goal is not only to reduce computational cost but also to improve training stability** (RMSNorm in a standard Pre-LN configuration is known to suffer from optimization instabilities in deep transformers. Simply approximating its variance (to speed it up) does not address this underlying stability issue.)
> - Motivated by your suggestion, **we additionally implemented a variant “RMSNorm-Approx”, where the variance in RMSNorm is approximated using our estimator.** We compare it with standard RMSNorm below:
>
> Tab. 2. RMSNorm vs RMSNorm-Approx
> | Llama-1B | Variance Approx. | PT Train Loss | PT Eval Loss | PT Eval PPL | Train steps per sec. |
> |:---:|:---:|:---:|:---:|:---:|:---:|
> | RMSNorm | X | **3.281** | **3.272** | **26.353** | 0.346 |
> | RMSNorm-Approx | O | 3.293 | 3.284 | 26.672 | **0.381** |
>
> | Llama-1B | Variance Approx. | Arc-e | PIQA | Hellaswag | OpenBookQA | Winogrande | MMLU | BoolQ | Avg. |
> |:---:|:---:|:---:|:---:|:---:|:---:|:---:|:---:|:---:|:---:|
> | RMSNorm | X | $\textbf{30.97}_{\pm{0.20}}$ | $\textbf{62.89}_{\pm{0.74}}$ | $\textbf{32.77}_{\pm{0.09}}$ | $32.32_{\pm{0.94}}$ | $50.54_{\pm{0.48}}$ | $\textbf{23.14}_{\pm{0.00}}$ | $\textbf{56.26}_{\pm{0.44}}$ | $\textbf{41.27}_{\pm{0.10}}$ |
> | RMSNorm-Approx | O | $31.15_{\pm{0.44}}$ | $61.98_{\pm{0.57}}$ | $31.93_{\pm{0.04}}$ | $\textbf{33.08}_{\pm{0.04}}$ | $\textbf{50.67}_{\pm{0.94}}$ | $22.95_{\pm{0.00}}$ | $46.32_{\pm{0.26}}$ | $39.72_{\pm{0.09}}$ |
>
> We observe that: **RMSNorm-Approx converges similarly** to RMSNorm in terms of loss, and **achieves higher training speed** (0.381 vs. 0.346 steps/sec). However, **its average downstream performance is slightly worse than standard RMSNorm**.
>
> We have also added this ablation in Tab. 5 in the revised manuscript.
>
> **Q3. Please include validation perplexity results of pretraining.**
>
> We appreciate the reviewer for emphasizing the importance of validation Perplexity (PPL). **In the revised manuscript, we have added validation PPL to all tables reporting pretraining results.**
>
> We believe this change improves the clarity and completeness of our empirical evaluation.
>
> **Q4. Hardware and measurement protocol details.**
>
> Thank you for pointing out the need for clearer experimental details. We have added a detailed description of our training setup in Appendix C.4 of the revised manuscript. The key information is:
>
> - Pretraining and SFT framework: Llama-Factory
> - Llama-1B GPU: NVIDIA RTX A6000
> - Llama-3B GPU: NVIDIA RTX PRO 6000 Blackwell
> - Parallel computation: FlashAttention-2 for efficient attention computation
> - Distributed training: DeepSpeed ZeRO-2
>
> Further configuration details are summarized in the revised appendix. We hope this makes our experiments easier to reproduce.

---

> ### Author Response · Authors · 2025-11-29
>
> **Q5. Add BoolQ results.**
>
> - We agree that including BoolQ is important for a fair comparison with LNS.
> - In the revised manuscript, we have added BoolQ scores to all evaluation tables that report downstream performance.

---

### Official Review · Reviewer_d4bW · 2025-10-31

**Soundness:** 4
**Presentation:** 3
**Contribution:** 3
**Rating:** 6
**Confidence:** 4

**Summary:**

The paper proposes the BHyT which replaces Pre-LN with "input bounded with probability guarantee + tanh" and only precisely calculates the variance once per Transformer block, with the rest using lightweight approximation.

**Strengths:**

- Combine the speed advantage of "unnormalized" with the stability of "bounded non-saturated" to explicitly avoid tanh saturation with probability limiting.
- The method is simple and effective.

**Weaknesses:**

- Only validated on 1B/3B, C4, and a small amount of SFT datasets; did not cover deeper layers or longer contexts. I understand that it is not realistic to do such a thing with limited resources, but perhaps training a narrow and deep model might be feasible?
- The "uniform" attention assumption is least tenable in which training stages/tasks? If the second moment of the real attention weights replaces the uniform assumption, what are the approximate errors and throughput losses?

**Questions:**

see weekness above

---

> ### Author Response · Authors · 2025-11-29
>
> **W1. Only validated on 1B/3B... perhaps training a narrow and deep model might be feasible?**
>
> We appreciate the suggestion regarding "narrow and deep" models.
>
> We designed our experiments to align with the Peri-LN [1] study, which utilized a 3.2B parameter model with up to 32 layers. Our **Llama-3B model (following Llama-3.2-3B architecture) has 28 layers**, **providing a comparable depth and parameter size to established stability research.**
>
> Furthermore, when we applied the best hyperparameters from the 1B model to the 3B model, BHyT still achieved the lowest pretrain loss among baselines. This suggests that BHyT's stability is not dependent on model size.
>
> To further address the concern about data scale, **we extended training for both 1B and 3B models to 20 Billion tokens on the C4 corpus.**
>
> Tab. 1. Llama-1B (Pretrained on 20B tokens only)
> | Method | PT Train Loss | PT Eval Loss | PT Eval PPL |
> |:---:|:---:|:---:|:---:|
> | Peri-LN | 2.870 | 2.846 | 17.22 |
> | BHyT | 2.828 | 2.802 | 16.47 |
>
> | Method | Arc-e | PIQA | Hellaswag | OpenBookQA | Winogrande | MMLU | BoolQ | Avg. |
> |:---:|:---:|:---:|:---:|:---:|:---:|:---:|:---:|:---:|
> | Peri-LN | $37.91_{\pm{0.23}}$ | $\textbf{69.16}_{\pm{0.33}}$ | $46.77_{\pm{0.10}}$ | $\textbf{31.36}_{\pm{0.82}}$ | $\textbf{50.89}_{\pm{0.96}}$ | $\textbf{27.71}_{\pm{0.00}}$ | $51.74_{\pm{0.63}}$ | $45.08_{\pm{0.09}}$ |
> | BHyT | $\textbf{40.98}_{\pm{0.43}}$ | $69.13_{\pm{0.34}}$ | $\textbf{49.00}_{\pm{0.07}}$ | $30.56_{\pm{0.73}}$ | $50.67_{\pm{0.75}}$ | $26.83_{\pm{0.00}}$ | $\textbf{55.86}_{\pm{0.38}}$ | $\textbf{46.15}_{\pm{0.23}}$ |
>
> Tab. 2. Llama-1B (Pretrained on 20B tokens & SFT)
> | Method | PT Train Loss | PT Eval Loss | PT Eval PPL | SFT Tr Loss | SFT Eval Loss |
> |:---:|:---:|:---:|:---:|:---:|:---:|
> | Peri-LN | 2.870 | 2.846 | 17.22 | $\textbf{2.522}_{\pm{0.011}}$ | $2.856_{\pm{0.107}}$ |
> | BHyT | **2.828** | **2.802** | **16.47** | $2.550_{\pm{0.011}}$ | $\textbf{2.825}_{\pm{0.112}}$ |
>
> | Method | Arc-e | PIQA | Hellaswag | OpenBookQA | Winogrande | MMLU | BoolQ | Avg. |
> |:---:|:---:|:---:|:---:|:---:|:---:|:---:|:---:|:---:|
> | Peri-LN | $47.90_{\pm{0.63}}$ | $\textbf{71.09}_{\pm{0.35}}$ | $47.79_{\pm{0.24}}$ | $\textbf{31.24}_{\pm{0.52}}$ | $\textbf{51.98}_{\pm{0.48}}$ | $26.68_{\pm{0.01}}$ | $48.74_{\pm{0.79}}$ | $46.44_{\pm{0.27}}$ |
> | BHyT | $\textbf{49.23}_{\pm{0.60}}$ | $71.02_{\pm{0.28}}$ | $\textbf{49.42}_{\pm{0.15}}$ | $30.72_{\pm{0.84}}$ | $51.49_{\pm{0.80}}$ | $\textbf{27.36}_{\pm{0.01}}$ | $\textbf{54.83}_{\pm{0.94}}$ | $\textbf{47.72}_{\pm{0.31}}$ |
>
> - BHyT consistently achieves lower pretraining loss and PPL than Peri-LN.
> - On downstream benchmarks, BHyT achieves higher average accuracy while remaining competitive or better on most individual tasks.
> - These results, described in detail in Table 9 in Appendix D.5 of the revised manuscript
>
> [1] Kim et al., Peri-LN: Revisiting Normalization Layer in the Transformer Architecture, ICML 2025
>
> **W2. The "uniform" attention assumption...**
>
> We thank the reviewer for raising this important conceptual point.
>
> We would like to clarify that **we approximate the variance of the attention module output, not the attention matrix itself.** The assumption that the attention matrix becomes uniform as the sequence length increases is derived from the theoretical analysis in the LNS [2].
>
> To validate that **this idealized assumption does not hurt the method**, we performed the **quantitative error analysis of the variance approximation reported in Tab. 7 of the Appendix D.2 in revied version.**
>
> The results show **very high correlation ($R^2$, Pearson $r$, Spearman $\rho \approx 0.99$) between the approximated and exact variances.**
>
> These findings suggest that **even though the uniform attention assumption is simplified, the resulting variance approximation remains very accurate across realistic training conditions.**
>
> [2] Sun et al., The Curse of Depth in Large Language Models, ArXiv 2025

---

> ### Author Response · Authors · 2025-12-03
>
> Since the 20B-token pretraining of the LLaMA-3B model was completed before the discussion period, we are sharing the results.
>
> Tab. 3. Llama-3B (Pretrained on 20B tokens only)
> | Method | PT Train Loss | PT Eval Loss | PT Wall time |
> |:---:|:---:|:---:|:---:|
> | Peri-LN | 2.811 | 2.812 | 238h |
> | BHyT | 2.756 | 2.760 | 171.4h |
>
> | Method | Arc-e | PIQA | Hellaswag | OpenBookQA | Winogrande | MMLU | BoolQ | Avg. |
> |:---:|:---:|:---:|:---:|:---:|:---:|:---:|:---:|:---:|
> | Peri-LN | $41.03_{\pm{0.56}}$ | $68.93_{\pm{0.67}}$ | $47.47_{\pm{0.21}}$ | $30.64_{\pm{0.99}}$ | $51.70_{\pm{0.26}}$ | $27.02_{\pm{0.00}}$ | $47.192_{\pm{0.74}}$ | $44.85_{\pm{0.27}}$ |
> | BHyT | $\textbf{45.93}_{\pm{0.46}}$ | $\textbf{70.09}_{\pm{0.14}}$ | $\textbf{50.97}_{\pm{0.14}}$ | $\textbf{31.32}_{\pm{1.00}}$ | $\textbf{52.12}_{\pm{0.72}}$ | $\textbf{27.62}_{\pm{0.00}}$ | $\textbf{55.69}_{\pm{0.86}}$ | $\textbf{47.68}_{\pm{0.5}}$ |
>
> Tab. 4. Llama-3B (Pretrained on 20B tokens & SFT)
>
> | Method | PT Train Loss | PT Eval Loss | PT Wall time | SFT Tr Loss | SFT Eval Loss |
> |:---:|:---:|:---:|:---:|:---:|:---:|
> | Peri-LN | 2.811 | 2.812 | 238h | $2.477_{\pm{0.011}}$ | $2.830_{\pm{0.10}}$ |
> | BHyT | 2.756 | 2.760 | 171.4h | $\textbf{2.468}_{\pm{0.011}}$ | $\textbf{2.764}_{\pm{0.10}}$ |
>
> | Method | Arc-e | PIQA | Hellaswag | OpenBookQA | Winogrande | MMLU | BoolQ | Avg. |
> |:---:|:---:|:---:|:---:|:---:|:---:|:---:|:---:|:---:|
> | Peri-LN | $50.19_{\pm{0.34}}$ | $70.93_{\pm{0.36}}$ | $48.39_{\pm{0.15}}$ | $\textbf{31.00}_{\pm{0.58}}$ | $51.98_{\pm{0.64}}$ | $26.36_{\pm{0.11}}$ | $42.43_{\pm{0.99}}$ | $45.90_{\pm{0.21}}$ |
> | BHyT | $\textbf{53.83}_{\pm{0.38}}$ | $\textbf{72.36}_{\pm{0.43}}$ | $\textbf{51.62}_{\pm{0.15}}$ | $30.60_{\pm{0.57}}$ | $\textbf{53.07}_{\pm{0.47}}$ | $\textbf{26.82}_{\pm{0.10}}$ | $\textbf{49.89}_{\pm{1.51}}$ | $\textbf{48.31}_{\pm{0.27}}$ |
>
> Even with a larger model size and more training tokens, BHyT continued to provide stable and faster training as well as higher evaluation performance compared to Peri-LN.
> These results described in detail in Tab. 4 in Sec. 4.5 of the revised manuscript.

---

### Official Review · Reviewer_iv9T · 2025-11-07

**Soundness:** 2
**Presentation:** 2
**Contribution:** 2
**Rating:** 4
**Confidence:** 4

**Summary:**

The paper introduces Bounded Hyperbolic Tanh (BHyT), a normalization-free alternative to Pre-Layer Normalization designed to improve both stability and efficiency in large language model training. It combines a tanh nonlinearity with an explicit, data-driven input bounding mechanism that constrains activations within a stable, non-saturating range, preventing variance explosion across depth.
BHyT theoretically guarantees gradient stability by bounding the Jacobian norm relative to RMSNorm and uses a lightweight variance approximation to reduce normalization overhead. Empirically, it aims to maintain training stability comparable to heavily normalized methods like Peri-LN while achieving higher computational efficiency. Overall, the paper positions BHyT as a practical replacement for Pre-LN, balancing stability and throughput for large-scale Transformer models.

**Strengths:**

- Clear motivation.
- Conceptually simple yet theoretically grounded design.
- Lightweight variance approximation for efficiency.
- Empirical evidence of improved stability.

**Weaknesses:**

- **Inadequate reporting and questionable generality of Peri-LN throughput results.** Figure 4(b) claims that Peri-LN achieves strong accuracy but suffers from the slowest throughput, positioning BHyT as the best trade-off. However, the paper does not specify the environment under which throughput was measured. All experiments were conducted in Llama-Factory rather than in standard large-scale frameworks (e.g., Megatron-Core or NeMo) that officially support Gemma-style Peri-LN. Without such metadata or a cross-framework check, it is unclear whether the reported slowdown reflects an intrinsic limitation of Peri-LN or merely framework-specific overhead.

- **Under-trained experimental regime and limited statistical reliability.** The 1B and 3B models were trained on only 1.64 B and 1.97 B tokens, respectively—orders of magnitude below standard compute-optimal ratios. These under-saturated runs make it difficult to assess convergence, generalization, or stability trends. No results are averaged over multiple seeds or accompanied by confidence intervals. Reported downstream improvements (Tables 1–4) therefore lack statistical robustness, and the unusually large MMLU jump (36.54 % for BHyT) raises concerns about configuration consistency and reproducibility.

- **Theory–practice gap and missing validation of assumptions.** Theoretical guarantees apply only to the idealized variant BHyT* with exact statistics and zero-mean inputs. The implemented BHyT uses an approximated variance under strong assumptions—uniform attention, Gaussianity, and linearized tanh—but the paper offers no quantitative error analysis of this approximation (e.g., deviation across layers or attention-entropy regimes). As a result, the claimed Jacobian-norm bound and variance-stability guarantees remain unverified in realistic conditions.

- **Missing baselines and scale comparisons.** At the 3B scale, BHyT is compared only with DyT, omitting RMSNorm, LNS, and Peri-LN under identical configurations. This prevents a fair evaluation of scalability and undermines the claim that BHyT preserves stability at greater depth.

**Questions:**

stated in the weaknesses section.

---

> ### Author Response · Authors · 2025-11-29
>
> **W1. Inadequate reporting and questionable generality of Peri-LN throughput results.**
>
> We thank the reviewer for this valuable feedback and fully agree that cross-framework verification is important to assess the true efficiency of Peri-LN and BHyT.
>
> - To isolate throughput from framework-specific overhead, we loaded the models using **Hugging Face Transformers**, and measured tokens-per-second (TPS) using the generate() method, **without using the Llama-Factory framework.**
>
> The following table reports TPS for Llama-1B with sequnce length set to 4096, input prompt length fixed to 512 tokens, and varying maximum generation length:
>
> Tab. 1. TPS comparison between Peri-LN and BHyT
> | TPS | 128 | 512 | 1024 |
> |:---:|:---:|:---:|:---:|
> | Peri-LN | 73.449 | 76.408 | 77.616 |
> | BHyT | **90.852** | **92.096** | **93.011** |
>
> As shown, **BHyT consistently achieves higher TPS than Peri-LN** even in this **framework-agnostic setting**. This confirms that the throughput gains of **BHyT are not an artifact of Llama-Factory,** but stem from its design.
>
> In the revised manuscript, we clarify the measurement setup and present more detailed speed comparisons in Fig. 4: Fig. 4(a) reports the training loss curves, showing that BHyT trains efficiently. Fig. 4(b) compares forward pass speed, showing that BHyT is as fast as DyT and significantly faster than Peri-LN.
>
> We believe these additional experiments and clarifications resolve the concern about the generality and reporting of throughput results.
>
> **W2. Under-trained experimental regime and limited statistical reliability.**
>
> We appreciate the careful assessment of our training setup and the request for stronger statistical evidence.
>
> In Peri-LN [1], the authors pretrain a ~1.5B-parameter model (excluding embeddings) on 30B tokens, which corresponds to roughly 20× the non-embedding parameter count.
>
> In additional experiment, we follow a similar policy:
> - We adopt Llama-3.2-1B and Llama-3.2-3B architectures. For Llama-3.2-1B, the number of parameters excluding embeddings is **approximately 1B.** Accordingly, we **pretrain on 20B tokens**, which corresponds to approximately 20× the model size, matching the Peri-LN regime in terms of tokens-per-parameter.
> - Due to time and resource limitations, we focused our extended experiments on Peri-LN and BHyT, as Peri-LN is the most relevant and strongest baseline for training stability.
> - For the 1B model, the extended experiments have finished, and we report the results in the revised manuscript. The 3B experiments are still in progress, and we plan to share those results with the reviewers before the discussion period ends.
>
> Tab. 1. Llama-1B (Pretrained on 20B tokens only)
> | Method | PT Train Loss | PT Eval Loss | PT Eval PPL |
> |:---:|:---:|:---:|:---:|
> | Peri-LN | 2.870 | 2.846 | 17.22 |
> | BHyT | 2.828 | 2.802 | 16.47 |
>
> | Method | Arc-e | PIQA | Hellaswag | OpenBookQA | Winogrande | MMLU | BoolQ | Avg. |
> |:---:|:---:|:---:|:---:|:---:|:---:|:---:|:---:|:---:|
> | Peri-LN | $37.91_{\pm{0.23}}$ | $\textbf{69.16}_{\pm{0.33}}$ | $46.77_{\pm{0.10}}$ | $\textbf{31.36}_{\pm{0.82}}$ | $\textbf{50.89}_{\pm{0.96}}$ | $\textbf{27.71}_{\pm{0.00}}$ | $51.74_{\pm{0.63}}$ | $45.08_{\pm{0.09}}$ |
> | BHyT | $\textbf{40.98}_{\pm{0.43}}$ | $69.13_{\pm{0.34}}$ | $\textbf{49.00}_{\pm{0.07}}$ | $30.56_{\pm{0.73}}$ | $50.67_{\pm{0.75}}$ | $26.83_{\pm{0.00}}$ | $\textbf{55.86}_{\pm{0.38}}$ | $\textbf{46.15}_{\pm{0.23}}$ |
>
> Tab. 2. Llama-1B (Pretrained on 20B tokens & SFT)
> | Method | PT Train Loss | PT Eval Loss | PT Eval PPL | SFT Tr Loss | SFT Eval Loss |
> |:---:|:---:|:---:|:---:|:---:|:---:|
> | Peri-LN | 2.870 | 2.846 | 17.22 | $\textbf{2.522}_{\pm{0.011}}$ | $2.856_{\pm{0.107}}$ |
> | BHyT | **2.828** | **2.802** | **16.47** | $2.550_{\pm{0.011}}$ | $\textbf{2.825}_{\pm{0.112}}$ |
>
> | Method | Arc-e | PIQA | Hellaswag | OpenBookQA | Winogrande | MMLU | BoolQ | Avg. |
> |:---:|:---:|:---:|:---:|:---:|:---:|:---:|:---:|:---:|
> | Peri-LN | $47.90_{\pm{0.63}}$ | $\textbf{71.09}_{\pm{0.35}}$ | $47.79_{\pm{0.24}}$ | $\textbf{31.24}_{\pm{0.52}}$ | $\textbf{51.98}_{\pm{0.48}}$ | $26.68_{\pm{0.01}}$ | $48.74_{\pm{0.79}}$ | $46.44_{\pm{0.27}}$ |
> | BHyT | $\textbf{49.23}_{\pm{0.60}}$ | $71.02_{\pm{0.28}}$ | $\textbf{49.42}_{\pm{0.15}}$ | $30.72_{\pm{0.84}}$ | $51.49_{\pm{0.80}}$ | $\textbf{27.36}_{\pm{0.01}}$ | $\textbf{54.83}_{\pm{0.94}}$ | $\textbf{47.72}_{\pm{0.31}}$ |
>
> - **BHyT consistently achieves lower pretraining loss and PPL than Peri-LN.**
> - On downstream benchmarks, **BHyT achieves higher average accuracy while remaining competitive or better on most individual tasks.**
> - These results, described in detail in Tab. 9 in Appendix D.5 of the revised manuscript
>
> [1] Kim et al., Peri-LN: Revisiting Normalization Layer in the Transformer Architecture, ICML 2025.

---

> ### Author Response · Authors · 2025-11-29
>
> **W3. Theory–practice gap and missing validation of assumptions.**
>
> We thank the reviewer for pointing out the importance of validating the variance approximation empirically.
>
> **To quantify the approximation error**, we conducted the following experiment: For five random seeds, we randomly sampled 100 documents from the C4 corpus, **computed the exact variance of the attention module output, and compared it with the approximated variance used by BHyT**.
>
> We report the **average discrepancy and similarity between the exact and approximated variance**, together with their standard deviations:
>
> Tab. 3.
> |        |                     |
> |--------|---------------------|
> |  RMSE  | $2.124_{\pm{5e-2}}$ |
> |  $R^2$ |  $0.99_{\pm{4e-5}}$ |
> |   $r$  |  $0.99_{\pm{1e-5}}$ |
> | $\rho$ |  $0.99_{\pm{5e-5}}$ |
>
> (Reported as Table 7 in the appendix of the revised manuscript.)
>
> The very high $R^2$ (the coefficient of determination), Pearson correlation $r$, and Spearman correlation $\rho$ indicate that the approximated variance is extremely close to the exact variance across layers and attention regimes.
>
> **These results empirically support our theoretical claim that BHyT’s variance stabilization.**
>
> We believe this quantitative analysis narrows the theory–practice gap and directly addresses the concern.
>
> Furthermore, to strengthen the theoretical foundation regarding stability, we have added Theorem 3.6 in the revised manuscript. This theorem proves that BHyT achieves a strictly lower layer-wise variance bound compared to LayerNorm Scaling (LNS) [2].
>
> [2] Sun et al., The Curse of Depth in Large Language Models, ArXiv 2025
>
> **W4. Missing baselines and scale comparisons.**
>
> We appreciate the reviewer’s suggestion to include more comprehensive baselines at the 3B scale.
>
> In the revised manuscript, **we have extended our 3B experiments to all baslines and BHyT.**
>
> Tab. 4. Llama-3B (Pretrain only)
> | Method | PT Train Loss | PT Eval Loss | PT Eval PPL |
> |:---:|:---:|:---:|:---:|
> | RMSNorm | 3.203 | 3.180 | 24.040 |
> | Peri-LN | 3.165 | 3.142 | 23.156 |
> | LNS | 3.160 | 3.139 | 23.091 |
> | DyT | 3.877 | 3.855 | 47.244 |
> | BHyT | **3.133** | **3.107** | **22.346** |
>
> | Method | Arc-e | PIQA | Hellaswag | OpenBookQA | Winogrande | MMLU | BoolQ | Avg. |
> |:---:|:---:|:---:|:---:|:---:|:---:|:---:|:---:|:---:|
> | RMSNorm | $31.01_{\pm{0.21}}$ | $\textbf{66.57}_{\pm{0.41}}$ | $\textbf{41.52}_{\pm{0.25}}$ | $\textit{33.92}_{\pm{0.33}}$ | $50.50_{\pm{0.78}}$ | $22.98_{\pm{0.00}}$ | $37.94_{\pm{0.12}}$ | $40.63_{\pm{0.18}}$ |
> | Peri-LN | $31.82_{\pm{0.38}}$ | $64.52_{\pm{0.24}}$ | $36.05_{\pm{0.11}}$ | $32.28_{\pm{0.59}}$ | $49.30_{\pm{0.52}}$ | $22.99_{\pm{0.00}}$ | $\textit{57.46}_{\pm{0.31}}$ | $\textit{42.06}_{\pm{0.25}}$ |
> | LNS | $\textbf{31.88}_{\pm{0.42}}$ | $64.68_{\pm{0.46}}$ | $36.25_{\pm{0.10}}$ | $32.45_{\pm{0.53}}$ | $\textit{51.18}_{\pm{0.75}}$ | $\textbf{23.59}_{\pm{0.00}}$ | $37.83_{\pm{0.08}}$ | $39.69_{\pm{0.11}}$ |
> | DyT | $27.69_{\pm{0.27}}$ | $59.20_{\pm{0.15}}$ | $25.96_{\pm{0.17}}$ | $31.85_{\pm{0.38}}$ | $49.17_{\pm{1.28}}$ | $23.08_{\pm{0.00}}$ | $48.05_{\pm{0.56}}$ | $37.86_{\pm{0.12}}$ |
> | BHyT | $\textit{31.84}_{\pm{0.15}}$ | $\textit{65.08}_{\pm{0.39}}$ | $\textit{36.48}_{\pm{0.04}}$ | $31.76_{\pm{0.33}}$ | $\textbf{51.62}_{\pm{0.66}}$ | $\textit{23.22}_{\pm{0.00}}$ | $\textbf{60.84}_{\pm{0.29}}$ | $\textbf{42.98}_{\pm{0.16}}$ |
>
> - **BHyT achieves the lowest pretraining loss and PPL among all baselines at the 3B scale.**
> - On downstream tasks, **BHyT achieves the best average performance**, while remaining competitive on individual benchmarks.
> - These 3B comparisons, reported as Table 1 and Table 2 in the revised manuscript, provide a fair and comprehensive evaluation of scalability and stability across all normalization strategies.

---

> ### Author Response · Authors · 2025-12-03
>
> Since the 20B-token pretraining of the LLaMA-3B model was completed before the discussion period, we are sharing the results.
>
> Tab. 5. Llama-3B (Pretrained on 20B tokens only)
> | Method | PT Train Loss | PT Eval Loss | PT Wall time |
> |:---:|:---:|:---:|:---:|
> | Peri-LN | 2.811 | 2.812 | 238h |
> | BHyT | 2.756 | 2.760 | 171.4h |
>
> | Method | Arc-e | PIQA | Hellaswag | OpenBookQA | Winogrande | MMLU | BoolQ | Avg. |
> |:---:|:---:|:---:|:---:|:---:|:---:|:---:|:---:|:---:|
> | Peri-LN | $41.03_{\pm{0.56}}$ | $68.93_{\pm{0.67}}$ | $47.47_{\pm{0.21}}$ | $30.64_{\pm{0.99}}$ | $51.70_{\pm{0.26}}$ | $27.02_{\pm{0.00}}$ | $47.192_{\pm{0.74}}$ | $44.85_{\pm{0.27}}$ |
> | BHyT | $\textbf{45.93}_{\pm{0.46}}$ | $\textbf{70.09}_{\pm{0.14}}$ | $\textbf{50.97}_{\pm{0.14}}$ | $\textbf{31.32}_{\pm{1.00}}$ | $\textbf{52.12}_{\pm{0.72}}$ | $\textbf{27.62}_{\pm{0.00}}$ | $\textbf{55.69}_{\pm{0.86}}$ | $\textbf{47.68}_{\pm{0.5}}$ |
>
> Tab. 6. Llama-3B (Pretrained on 20B tokens & SFT)
>
> | Method | PT Train Loss | PT Eval Loss | PT Wall time | SFT Tr Loss | SFT Eval Loss |
> |:---:|:---:|:---:|:---:|:---:|:---:|
> | Peri-LN | 2.811 | 2.812 | 238h | $2.477_{\pm{0.011}}$ | $2.830_{\pm{0.10}}$ |
> | BHyT | 2.756 | 2.760 | 171.4h | $\textbf{2.468}_{\pm{0.011}}$ | $\textbf{2.764}_{\pm{0.10}}$ |
>
> | Method | Arc-e | PIQA | Hellaswag | OpenBookQA | Winogrande | MMLU | BoolQ | Avg. |
> |:---:|:---:|:---:|:---:|:---:|:---:|:---:|:---:|:---:|
> | Peri-LN | $50.19_{\pm{0.34}}$ | $70.93_{\pm{0.36}}$ | $48.39_{\pm{0.15}}$ | $\textbf{31.00}_{\pm{0.58}}$ | $51.98_{\pm{0.64}}$ | $26.36_{\pm{0.11}}$ | $42.43_{\pm{0.99}}$ | $45.90_{\pm{0.21}}$ |
> | BHyT | $\textbf{53.83}_{\pm{0.38}}$ | $\textbf{72.36}_{\pm{0.43}}$ | $\textbf{51.62}_{\pm{0.15}}$ | $30.60_{\pm{0.57}}$ | $\textbf{53.07}_{\pm{0.47}}$ | $\textbf{26.82}_{\pm{0.10}}$ | $\textbf{49.89}_{\pm{1.51}}$ | $\textbf{48.31}_{\pm{0.27}}$ |
>
> Even with a larger model size and more training tokens, BHyT continued to provide stable and faster training as well as higher evaluation performance compared to Peri-LN.
> These results, described in detail in Tab. 4 in Sec. 4.5 of the revised manuscript.

---

### Author Response · Authors · 2025-11-29
**Global response**

We thank for the careful and constructive feedback from all reviewers. All of the reviewers' comments have significantly helped us strengthen the empirical rigor and theoretical clarity of our work.

We summarize our major updates organized by common themes.

1. Scalability and Convergence: Extended pretraining & 3B validation [Relates to: iv9T-W2, iv9T-W4, d4bW-W1, fZC6-W3, fZC6-W4]
    - To address concerns regarding data scale and model depth, we significantly expanded our experiments:
    - 20B tokens training: **Pretraining on 20B tokens** confirms that **BHyT consistently achieves lower evaluation loss and higher downstream accuracy than Peri-LN for both the 1B and 3B size models.**
        - For the LLaMA-3B model, BHyT achieves about 28% faster pretraining and on average about 5% higher benchmark evaluation performance.
    - **Llama-3B** (Full Comparison): In full comparisons against all baselines (RMSNorm, Peri-LN, LNS, DyT), **BHyT achieved the lowest pretraining loss (3.133) and best average downstream performance (42.98%).**
    - Conclusion: These results demonstrate that **BHyT’s stability and performance benefits are robust across larger scales and extended training regimes.**

2. Efficiency Validation: Cross-Framework Check & Time-to-Target [Relates to: iv9T-W1, fZC6-W1, fZC6-W2]
    - We verified that BHyT’s efficiency is intrinsic and not an implementation artifact:
    - **Framework-Agnostic Throughput**: Measured via native Hugging Face Transformers, **BHyT maintains a significant throughput advantage (~93 (BHyT) vs. ~77 (Peri-LN) Tokens per second (TPS)) over Peri-LN.**
    - **Time-to-Target Analysis**: In a compute-constrained setting, BHyT reached the **target loss ~9.4 hours faster (33.22h vs. 42.62h) than Peri-LN.**
    - Conclusion: **BHyT** significantly outperforms Peri-LN in "real-world" **training efficiency** by eliminating the algorithmic overhead of repeated normalizations.

3. Theoretical Validity: Variance Approximation Analysis [Relates to: iv9T-W3, d4bW-W2, 5S5H-Q1, 5S5H-Q2]
    - Theoretical variance bound (Theorem 3.6): **We derived a finite-depth variance bound proving that BHyT achieves strictly lower variance growth than LayerNorm Scaling (LNS)** under realistic depth conditions, **providing a guarantee for its stability superiority.**
    - We empirically validated the "uniform attention" assumption and variance approximation:
        - Error Analysis: The **approximated variance shows a near-perfect correlation with the exact variance** on the C4 corpus.
        - Ablation (BHyT$^\ast$ vs. BHyT): Comparison with the exact-variance variant (BHyT$^\ast$) confirms that **our approximation yields a negligible performance difference while providing a clear boost in training speed** (0.385 (BHyT) vs. 0.335 (BHyT$^\ast$) steps/sec).

4. Completeness and Reproducibility [Relates to: 5S5H-Q3, 5S5H-Q4, 5S5H-Q5]
    - Metrics: Validation Perplexity (PPL) and BoolQ results have been added to all relevant tables.
    - Details: Detailed hardware specifications and training configurations (FlashAttention-2, DeepSpeed ZeRO-2) are now fully documented in Appendix C.4.

We believe these extensive revisions and new experimental evidence comprehensively address the reviewers' concerns. We are confident that BHyT stands as a robust, scalable, and efficient alternative for training stable LLMs.

We sincerely appreciate the time and effort of the reviewers.

---

### Meta-Review · Area_Chair_qHdR · 2026-01-06

**Summary:**

Overall, this seems to be a solid paper. Reviewers commented on it being clear, simple, solid and supported by empirical evidence. The authors' rebuttals were comprehensive, with quite a number of new results (e.g addressing model sizes, # training tokens, etc. etc.).

Unfortunately, the key deciding factor for this AC was that none of the reviewers was particularly enthusiastic or strongly supportive of this work, both qualitatively or quantitatively. There was little to no mention of strong novelty or significance.

For a highly-selective conference like ICLR, this paper, while relatively solid, does not have a stronger case for acceptance in comparison to other submissions.

I encourage the authors to take heart. Subsequent submissions to other venues may be better received or more suitable. Also, for a work such as this, "the proof is in the pudding"... if BHyT really works well, a combination of arxiv, blog posts or social media may convince others to try your method, and the ultimate test is whether others use it because it works well for them.

**Reviewer Concerns:**

Unfortunately, there were no reviewer responses to the rebuttals. Overall, I would hazard that most concerns were addressed partially or wholly by the rebuttal. That said, as mentioned above, there was no particularly strong support for this paper in terms of its strengths, so this was the deciding factor for this paper, rather than specific concerns.

**Reviewer Scores:**

Most reviewer concerns were relatively minor, and were often clarifications or requests for more results (e.g. for better gauge of robustness or generality). I would think most reviewers would have a 50/50 chance of raising their scores, as the rebuttals were fairly comprehensive, with quite a few new results.

One possible exception might be Reviewer iv9T, who had a number of strong concerns. While the authors responded in full, whether the reviewer might be convinced is less likely, based on the strong first review impression.

---

### Decision · Program_Chairs · 2026-01-26

Reject